# FLOWNAR: Scalable Streaming Narration for Long-Form Videos

**Zeyun Zhong** [1 2]    **Manuel Martin** [2]    **Chengzhi Wu** [1]    **David Schneider** [1]
**Frederik Diederichs** [2]    **Juergen Gall** [3 4]    **Juergen Beyerer** [1 2]

## Abstract

Recent Large Multimodal Models (LMMs), primarily designed for offline settings, are ill-suited for the dynamic requirements of streaming video. While recent online adaptations improve real-time processing, they still face critical scalability challenges, with resource demands typically growing at least linearly with video duration. To overcome this bottleneck, we propose FLOWNAR, a novel framework for scalable streaming video narration. The core of FLOWNAR is a dynamic context management strategy for historical visual context removal, combined with our CLAM (Cross Linear Attentive Memory) module for streaming visual history retention, ensuring bounded visual memory usage and computational complexity, crucial for efficient streaming. We also introduce a realistic self-conditioned evaluation protocol and complementary evaluation metrics to assess streaming narration models under deployment-like conditions. Experiments on the Ego4D, EgoExo4D, and EpicKitchens100 datasets demonstrate that FLOWNAR substantially improves narration quality over strong baselines while being highly efficient, supporting processing of $10\times$ longer videos and achieving $3\times$ higher throughput (FPS). The code is available at https://github.com/zeyun-zhong/FlowNar.

## 1. Introduction

The rapid evolution of video content processing has necessitated the development of models capable of operating in real-time, streaming environments. While the domain of video understanding has been significantly enriched by large multimodal models (LMMs) (Alayrac et al., 2022; Li et al., 2023; Liu et al., 2023a; Zhu et al., 2024; Ouyang et al., 2022; Song et al., 2024), these systems are predominantly designed for offline use, making them ill-suited for the dynamic, continuous nature of streaming video where immediate interpretation is crucial. This limitation highlights the need for effective online models that process video data sequentially as it arrives.

Pioneering efforts in online video understanding with LMMs, such as Videollm-online (Chen et al., 2024), demonstrated the feasibility of generating frame-aligned narrations for continuous video input. However, a critical bottleneck persists: due to continuous visual context accumulation, these online approaches exhibit resource demands scaling at least linearly with video duration. Consequently, they often exceed common VRAM capacities, as illustrated in Figure 1 (middle), or their processing speed significantly degrades as more frames are processed (right). This growing complexity hinders their application to the increasingly relevant domain of long-form video analysis.

To tackle these scaling resource demands, we propose FLOWNAR, a novel framework for scalable streaming video narration built on two key principles: (1) a dynamic context management (DCM) strategy for robust and efficient inference, and (2) a Cross Linear Attentive Memory (CLAM) module to retain crucial visual historical information. Our DCM primarily involves pruning the detailed visual KV cache after each narration segment. This provides a dual benefit: it ensures the LLM's active visual context does not grow unboundedly with past frame features, and it reduces error propagation that can arise from conditioning on potentially misaligned history. To train our model effectively under this paradigm, where it cannot directly attend to detailed past frames, we employ a specialized attention masking strategy during training, forcing it to learn to utilize current visual cues and available narration history optimally.

However, while aggressive pruning is vital for scalability, simply discarding all visual history or retaining only a fixed window of the most recent visual frames would lead to a significant loss of long-term visual information, compromising the quality and accuracy of subsequent narration. We experimentally demonstrate this limitation in our work. To preserve essential historical visual information, we re-

---

[1]Karlsruhe Institute of Technology (KIT) [2]Fraunhofer IOSB [3]Lamarr Institute for Machine Learning and Artificial Intelligence [4]University of Bonn. Correspondence to: Zeyun Zhong <zeyun.zhong@kit.edu>.

*Proceedings of the $43^{rd}$ International Conference on Machine Learning*, Seoul, South Korea. PMLR 306, 2026. Copyright 2026 by the author(s).

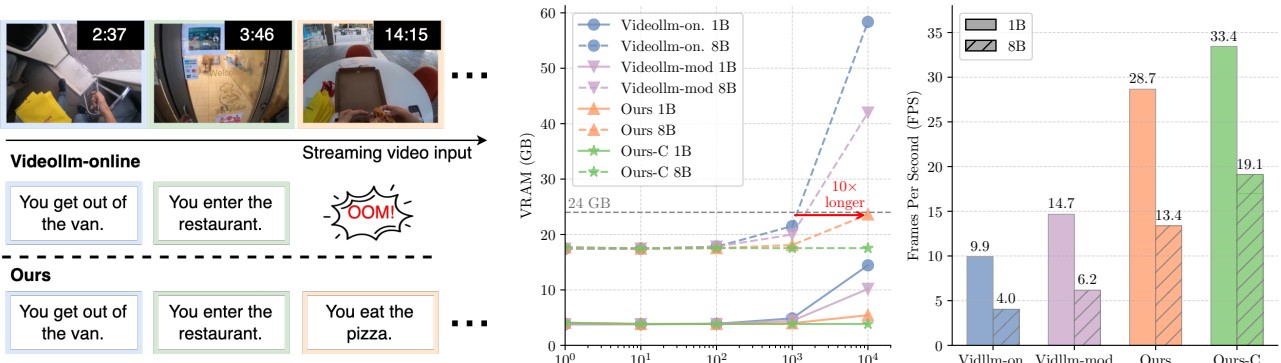

*Figure 1.* Efficiency and scalability comparison of FLOWNAR, Videollm-online (Chen et al., 2024), and Videollm-mod (Wu et al., 2024). (Left) Example streaming outputs: Videollm-online can encounter out of memory (OOM) errors while FLOWNAR continues. (Middle) VRAM usage vs. processed frames (log scale): memory usage for both baselines grows rapidly and exceeds a typical GPU limit (24GB, dashed line), whereas FLOWNAR variants remain compact, supporting 10× longer videos relative to Videollm-online. (Right) Throughput (FPS) on an H100 measured over 10K frames: FLOWNAR achieves roughly 3× higher FPS than Videollm-online.

formulate linear attention (Katharopoulos et al., 2020) as a streaming visual compressor (CLAM). It iteratively extracts relevant visual information from processed segments into a fixed-size set of memory tokens. These tokens serve as the condensed summary of past visual frames passed to subsequent segments, offering constant memory usage and per-step computational complexity for historical visual frames. To achieve scalability for arbitrarily long videos, our FLOWNAR-C variant (where 'C' denotes Constant total memory) further manages textual context by retaining only the last-$k$ generated narrations, ensuring constant memory usage for both visual and narration history.

Furthermore, to evaluate streaming narration models under deployment-like conditions, we move beyond evaluations that rely on ground-truth historical narrations (Chen et al., 2024), which can mask compounding errors and overestimate real-world performance. Instead, we adopt a fully self-conditioned protocol in which the model's predictions are conditioned on its own previously generated narrations. We also introduce a suite of evaluation measures and a *first-align-then-evaluate* procedure that aligns predicted narrations to ground-truth segments post-hoc before computing evaluation metrics. Our extensive testing on three challenging long-form video datasets (Ego4D (Grauman et al., 2022), EgoExo4D (Grauman et al., 2024), EK100 (Damen et al., 2022)) demonstrates the robustness of our dynamic context management in mitigating such error propagation from misaligned historical context, and the effectiveness of our streaming visual compressor in providing beneficial visual summaries. In summary, our main contributions are:

- A scalable streaming LMM framework, FLOWNAR, employing dynamic context management to ensure bounded memory and computational complexity while reducing error propagation.

- A streaming memory module, CLAM, for effective summary of visual history with constant per-step computation and memory cost, essential for streaming applications.

- A realistic self-conditioned protocol, together with complementary evaluation metrics, for deployment-like assessment of narration performance.

- Experiments demonstrating FLOWNAR's significant narration improvements over baselines, especially under the self-conditioned protocol, with state-of-the-art efficiency.

## 2. Related work

**LMMs for online video understanding.** Large multimodal models (LMMs) (Alayrac et al., 2022; Li et al., 2023; Liu et al., 2023a) have significantly advanced multimodal comprehension, addressing various video understanding benchmarks, including action recognition (Zhao et al., 2023), temporal action localization (Liu et al., 2024), and video dialogue/question answering (Li et al., 2025a; Song et al., 2024; Lin et al., 2024). However, these models typically analyze entire videos *offline*, limiting their use in real-time applications like AR or autonomous driving. Recent work has begun to adapt LMMs to streaming scenarios. Videollm-online (Chen et al., 2024) proposed an LLM-based online narrator and its efficient variant (Wu et al., 2024) reduces intermediate visual computation, supporting 1.7× longer videos. However, these methods do not bound visual memory growth for long videos. In contrast, our approach combines dynamic context management with a streaming memory module to maintain near-constant visual-context complexity, empirically supporting 10× longer videos for processing in our experiments. We also introduce a realistic self-conditioned evaluation protocol and tailored metrics to better assess deployment-like behavior. Beyond the directly comparable baselines, ProVideLLM (Chat-

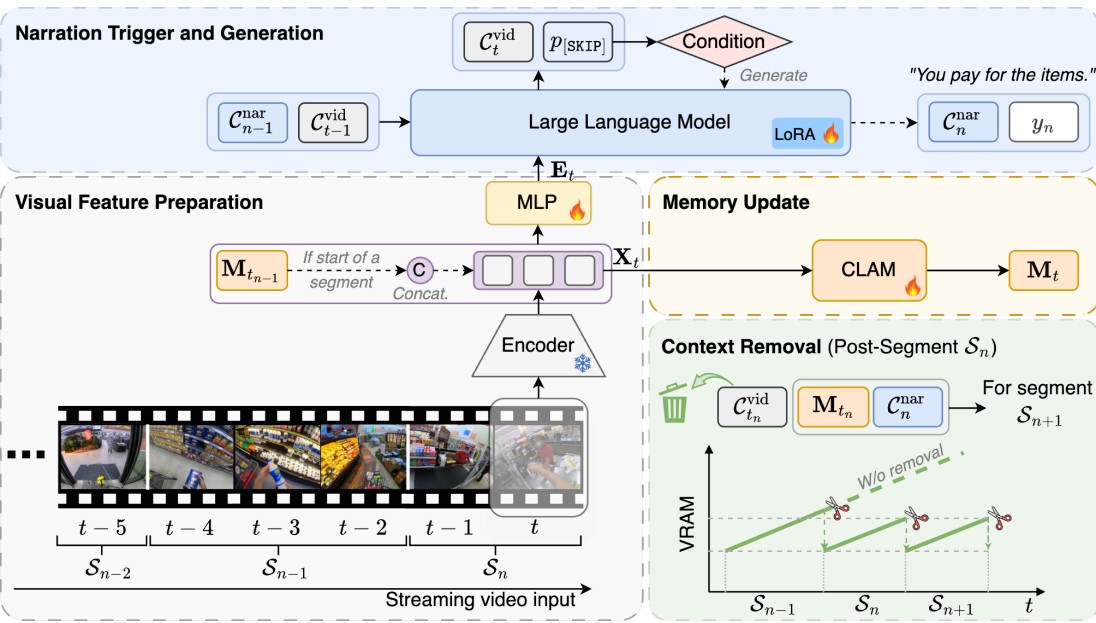

*Figure 2.* Overview of FLOWNAR. Streaming video frames $\mathbf{v}_t$ are encoded and projected, incorporating past memory $\mathbf{M}_{t_{n-1}}$ at segment start, to produce features $\mathbf{E}_t$. Our memory module (CLAM) updates memory tokens $\mathbf{M}_t$ using $\mathbf{X}_t$. The LLM processes $\mathbf{E}_t$ conditioned on cached contexts $(\mathcal{C}^{\mathrm{vid}}_{t-1}, \mathcal{C}^{\mathrm{nar}}_{n-1})$ to trigger the generation of a narration $y_n$. Post-generation, the updated memory $\mathbf{M}_{t_n}$ and narration context $\mathcal{C}^{\mathrm{nar}}_n$ are carried forward for processing the next segment, while the visual cache $\mathcal{C}^{\mathrm{vid}}_{t_n}$ is discarded to ensure bounded visual memory usage. Dashed arrows denote conditioned flow. Icons indicate trainable ( 🔥 ) vs. frozen ( ❄ ) modules.

terjee et al., 2025) targets the related but distinct task of memory-efficient step recognition over pre-segmented clips, rather than autonomous narration of an unsegmented stream. Meanwhile, another line of streaming video LMMs (Di et al., 2025; Yang et al., 2025; Zeng et al., 2025) addresses *query-triggered* question answering; in contrast, FLOWNAR performs streaming narration by autonomously deciding when to generate open-vocabulary descriptions without any external queries.

**Dynamic context management in LLMs.** Efficiently managing the Key-Value (KV) cache is critical for autoregressive LLM inference over long sequences (Shi et al., 2024). Prior strategies primarily target the textual KV cache. Sliding window attention (Beltagy et al., 2020; Jiang et al., 2023) offers simplicity by retaining only the KV pairs for a fixed window of recent tokens. More sophisticated cache eviction techniques selectively discard less relevant KV pairs based on attention scores (Xiao et al., 2024; Han et al., 2024; Liu et al., 2023b), or sparsification (Tang et al., 2025; Yao et al., 2024; Devoto et al., 2024), aiming to preserve crucial long-range information under a memory budget. Moving to the multi-modal regime, other works also address efficient visual history management. MovieChat (Song et al., 2024) merges similar visual tokens, while Zhou et al. (2024) apply online K-Means clustering to frame features. Several multimodal approaches compress or aggregate visual tokens but still allow memory to grow over time (Qian et al., 2024;

2025). By contrast, we introduce a neural streaming visual memory that incrementally compresses past visual frames into a fixed-size token bank and combine it with an explicit pruning policy (DCM), thereby bounding the memory of the cached visual state and reducing error propagation in autoregressive narration. A more detailed literature summary is provided in Appendix D.

## 3. Scalable streaming video narration

Our framework for scalable streaming video narration continuously processes long-form videos, deciding both *when* to generate a narration for an ongoing segment and *what* that narration should be. Given an untrimmed video represented as a sequence of frames $\mathbf{V} = \{\mathbf{v}_t\}_{t=1}^{T}$, where $T$ can be arbitrarily large, our objective is to process the video online, conceptually dividing it into segments $\mathcal{S}_n$. For each segment containing frames $\{\mathbf{v}_t\}_{t=t_{n-1}+1}^{t_n}$, the goal is to generate a corresponding textual narration $y_n$, producing the final output sequence $\Psi = \{(t_n, y_n)\}_{n=1}^{N}$. The generation operates in a streaming fashion, recursively producing each narration $(t_n, y_n)$ conditioned on the dynamically maintained context available at time $t_n$ (see Fig. 2). This context comprises two components: visual context $\mathcal{C}^{\mathrm{vid}}_t$ and narration context $\mathcal{C}^{\mathrm{nar}}_n$. The narration context $\mathcal{C}^{\mathrm{nar}}_n$ stores relevant information derived from previously generated narrations $\{y_j\}_{j=1}^{n-1}$ and is updated when a new narration $y_n$ is produced.

The visual context $\mathcal{C}_t^{\text{vid}}$ provides the necessary visual information up to the current time $t$. Naively caching context for all historical frames leads to undesirable memory and computational demands that grow at least linearly with video duration. While strategies such as discarding historical visual information entirely or only retaining a very recent window can be applied, they would lead to information loss from the more distant past. To overcome this, we propose a novel neural memory design to iteratively compress historical visual information into a fixed-size set of memory tokens, allowing the model to retain information from distant past. Our visual context thus distinctively combines two types of information: (1) a bounded summary of the preceding visual history (covering frames up to the end of the previous segment, $t_{n-1}$), maintained by our Cross Linear Attentive Memory (CLAM), and (2) the detailed, uncompressed frame information from the current segment $\mathcal{S}_n$. This compositional structure allows the model to leverage both a compressed representation of the long-term past and the fine-grained details of the recent visual input.

### 3.1. Visual feature extraction

We process each incoming frame $\mathbf{v}_t$ independently using a frozen pre-trained visual encoder (*e.g.*, SigLIP (Zhai et al., 2023)). From the encoder's output, we retain both the global CLS token and spatial information derived from a downsampled set of the patch tokens, following Wu et al. (2024). These selected tokens are combined to form the final frame representation $\mathbf{X}_t \in \mathbb{R}^{J \times D}$, which consists of $J$ tokens per frame, each with dimension $D$. An MLP (multi-layer perceptron) is used to map these representations into language feature space with dimension $D_{\text{lm}}$ for further processing in the LLM, $\mathbf{E}_t \in \mathbb{R}^{J \times D_{\text{lm}}}$.

### 3.2. Narration trigger and generation

Our framework utilizes a large language model (LLM, *e.g.*, Llama (Meta, 2024)) for both deciding *when* to narrate and *what* to narrate, processing incoming visual frame features ($\mathbf{E}_t$) alongside the evolving contexts. We denote the visual context after processing frame $t$ as $\mathcal{C}_t^{\text{vid}}$ (updated framewise) and the narration context after generating the $n^{\text{th}}$ narration $y_n$ as $\mathcal{C}_n^{\text{nar}}$ (updated segmentwise). These context states $\mathcal{C}_t^{\text{vid}}$ and $\mathcal{C}_n^{\text{nar}}$ correspond to the accumulated Key-Value (KV) pairs from visual and past narration tokens, respectively, within the LLM's attention layers. The system aims to identify segment endpoints $t_n$ online and generate $y_n$.

**Narration trigger.** To determine when to generate a narration, we evaluate the probability of predicting a special [SKIP] token at each potential decision point $t$, adapting the mechanism from Chen et al. (2024). This probability is conditioned on the current frame's features $\mathbf{E}_t$, the visual context up to the previous frame $\mathcal{C}_{t-1}^{\text{vid}}$, and the narration

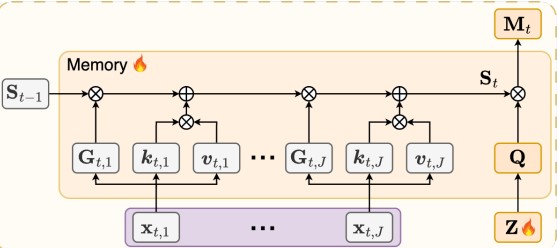

*Figure 3.* CLAM mechanism. A gated recurrent state update processes frame tokens $\{\mathbf{x}_{t,j}\}$ sequentially to update state $\mathbf{S}_t$ from $\mathbf{S}_{t-1}$. Learnable query vectors $\mathbf{Z}$ generate queries $\mathbf{Q}$ that read out fixed-size memory tokens $\mathbf{M}_t$ from the final state.

history from the last completed segment $\mathcal{C}_{n-1}^{\text{nar}}$. If this probability does not exceed an active threshold $\theta$:

$$p(\texttt{[SKIP]} \mid \mathbf{E}_t, \mathcal{C}_{t-1}^{\text{vid}}, \mathcal{C}_{n-1}^{\text{nar}}) \leq \theta, \qquad (1)$$

then the current time $t$ is designated as the $n^{\text{th}}$ narration endpoint, $t_n = t$, and generation proceeds. Otherwise ($p(\texttt{[SKIP]}) > \theta$), no narration is generated, the visual context is simply updated to $\mathcal{C}_t^{\text{vid}}$, and the system advances to frame $t + 1$. However, a single static threshold (Chen et al., 2024) can cause undesirable behaviors: bursts (multiple triggers in rapid succession) or long silences (extended gaps with no triggers). To mitigate both, we use a dynamic two-threshold strategy, using a main threshold $\theta$ by default, but we switch to a lower threshold $\theta_{\text{low}}$ (discouraging narration) for a short period after each narration $y_n$ is generated, preventing excessively rapid triggers. We provide an ablation study on this design in Section 4.3.

**Narration generation.** Once triggered at time $t_n$, the LLM autoregressively generates the $n^{\text{th}}$ narration $y_n = (w_{n,1}, \dots, w_{n,L_n})$. The generation is conditioned on the final visual context $\mathcal{C}_{t_n}^{\text{vid}}$ (which incorporates the update from frame $t_n$) and the relevant preceding narration context $\mathcal{C}_{n-1}^{\text{nar}}$. The probability for each token $w_{n,i}$ is thus modeled as $p(w_{n,i} \mid \mathcal{C}_{t_n}^{\text{vid}}, \mathcal{C}_{n-1}^{\text{nar}}, w_{n,<i})$. Following the generation of $y_n$, the narration context is updated to $\mathcal{C}_n^{\text{nar}}$.

### 3.3. Cross linear attentive memory (CLAM)

Naively caching KV pairs for all historical frames scales memory and compute at least linearly with video duration $T$, rendering long-form processing impractical. To address this, we introduce a neural visual memory that compresses previously observed frames into a fixed-size set of $M$ memory tokens, denoted $\mathbf{M}_t \in \mathbb{R}^{M \times D}$, where $M$ is much smaller than the number of processed frames. This compact representation serves as an efficient summary of the visual past, drastically reducing the memory footprint and enabling faster access to historical context.

Targeting scalable streaming with (1) constant memory complexity, (2) constant per-step computational complex-

ity during inference, and (3) high training parallelizability, we present CLAM (Fig. 3), which recasts linear attention (Katharopoulos et al., 2020) as a cross-attention-based visual compressor tailored to streaming video. We maintain a recurrent state matrix $\mathbf{S}_t \in \mathbb{R}^{D \times D}$ representing accumulated information up to the frame $t$. This state is updated from the previous frame's state $\mathbf{S}_{t-1}$ by iteratively processing each of the $J$ tokens $\mathbf{x}_{t,j}$ within the incoming frame representation $\mathbf{X}_t$. For each token $\mathbf{x}_{t,j}$, we derive its corresponding key $\mathbf{k}_{t,j}$, value $\mathbf{v}_{t,j}$, and a decay gate $\mathbf{G}_{t,j} \in \mathbb{R}^{D \times D}$ whose values are in the range (0, 1) to control history retention. The update employs a gated recurrence, applied sequentially across tokens within the frame ($j = 1, \ldots, J$), conceptually similar to Yang et al. (2024):

$$\mathbf{S}_{t,j} = \mathbf{G}_{t,j} \odot \mathbf{S}_{t,j-1} + \mathbf{k}_{t,j}^\top \mathbf{v}_{t,j}, \qquad (2)$$

where $\odot$ denotes element-wise multiplication, $\mathbf{S}_{t,0} = \mathbf{S}_{t-1}$, and the final state for the frame is $\mathbf{S}_t = \mathbf{S}_{t,J}$. Following the analysis of Zhong et al. (2025), the $D \times D$ recurrent state can store $O(D)$ independent key–value associations, which we find empirically sufficient for the long-form narration regime studied here (Sec. 4.3, Table 6).

As shown in Fig. 3, we obtain a fixed set of memory tokens $\mathbf{M}_t$ from the final state $\mathbf{S}_t$ by using $M$ learnable vectors $\mathbf{Z} \in \mathbb{R}^{M \times D}$. These vectors are transformed via a linear projection into query matrices $\mathbf{Q}$, which interact with the compact state $\mathbf{S}_t$ to produce the updated memory tokens $\mathbf{M}_t = \mathbf{Q}\mathbf{S}_t \in \mathbb{R}^{M \times D}$. This *cross linear attentive* design fulfills all three key properties essential for scalable streaming. The memory tokens computed at the end of a segment $\mathcal{S}_{n-1}$, denoted $\mathbf{M}_{t_{n-1}}$, are prepended once to the frame embeddings when processing the next segment $\mathcal{S}_n$ (see Fig. 2), providing the LLM with a condensed summary of the distant past. The superiority of CLAM over naive history management (such as simple last-$k$ frames or discarding visual history entirely) and other recent online compression mechanisms is experimentally demonstrated in Section 4.3.

## 3.4. Streaming narration with DCM

During streaming inference, FlowNar operates recursively to generate timestamped narrations $\Psi = \{(t_n, y_n)\}$, dynamically managing visual and narration contexts. The detailed step-by-step procedure, integrating memory updates, triggering, generation, and context management, is outlined in Algorithm 1. The core loop involves updating the visual memory state $\mathbf{S}_t$ and tokens $\mathbf{M}_t$ (Sec. 3.3), and evaluating the narration trigger condition (Sec. 3.2, Eq. 1). When a narration $y_n$ is triggered and generated at endpoint $t_n$, the narration context $\mathcal{C}_n^{\text{nar}}$ is updated accordingly. While theoretically the visual cache for the current segment ($\mathcal{C}_t^{\text{vid}}$) could grow indefinitely if no narration is triggered, our dynamic two-threshold trigger (Sec. 3.2) empirically ensures timely narration endpoints $t_n$. This enables periodic context

---

**Algorithm 1** Self-conditioned streaming narration.

**Input:** Frame tokens $\{\mathbf{X}_t\}_{t=1}^T$, trigger threshold $\theta$
**Output:** Timestamped narrations $\Psi$

1:  $(\Psi, \mathcal{C}_0^{\text{vid}}, \mathcal{C}_0^{\text{nar}}) \leftarrow (\emptyset, \emptyset, \emptyset)$  // Init output list & contexts
2:  $n \leftarrow 1$; $b_{\text{new}} \leftarrow$ False  // Init segment index & flag
3:  $\mathbf{S}_0 \leftarrow \mathbf{0}$  // Init memory state
4:  **for** $t = 1$ **to** $T$ **do**
5:  $\quad \mathbf{M}_t, \mathbf{S}_t \leftarrow \text{CLAM}(\mathbf{X}_t, \mathbf{S}_{t-1})$  // Update memory
6:  $\quad$ **if** $b_{\text{new}}$ **then**  // Prepend memory for new segment
7:  $\quad\quad \mathbf{X}_t \leftarrow [\mathbf{M}_{t_{n-1}} ; \mathbf{X}_t]$; $b_{\text{new}} \leftarrow$ False
8:  $\quad$ **end if**
9:  $\quad \mathbf{E}_t \leftarrow \text{MLP}(\mathbf{X}_t)$  // Embedding alignment
10: $\quad p_{\texttt{[SKIP]}}, \mathcal{C}_t^{\text{vid}} \leftarrow \text{LLM}(\mathbf{E}_t, \mathcal{C}_{t-1}^{\text{vid}}, \mathcal{C}_{n-1}^{\text{nar}})$  // Processing
11: $\quad$ **if** $p_{\texttt{[SKIP]}} \leq \theta$ **then**  // Trigger condition met
12: $\quad\quad t_n \leftarrow t$  // Update segment endpoint
13: $\quad\quad y_n, \mathcal{C}_n^{\text{nar}} \leftarrow \text{LLM}(\mathcal{C}_{t_n}^{\text{vid}}, \mathcal{C}_{n-1}^{\text{nar}})$  // Generation
14: $\quad\quad \Psi \leftarrow \Psi \cup \{(t, y_n)\}$  // Append result
15: $\quad\quad \mathbf{M}_{t_n} \leftarrow \mathbf{M}_t$  // Update memory for new segment
16: $\quad\quad \mathcal{C}_t^{\text{vid}} \leftarrow \emptyset$  // Discard visual context
17: $\quad\quad n \leftarrow n + 1$; $b_{\text{new}} \leftarrow$ True
18: $\quad$ **end if**
19: **end for**

---

removal, maintaining stable visual memory usage.

A crucial step for scalable inference is the visual context removal performed after each narration generation (Line 16 of Alg. 1, and bottom right of Fig. 2). We explicitly clear the visual KV cache ($\mathcal{C}_t^{\text{vid}} \leftarrow \emptyset$), removing the cached keys and values corresponding to the raw frame features $\{\mathbf{X}_t\}_{t \in \mathcal{S}_n}$ used within the just-completed segment $\mathcal{S}_n$, as well as the memory tokens $\mathbf{M}_{t_{n-1}}$ prepended at the start of that segment. This removal enforces reliance on the newly computed memory tokens $\mathbf{M}_{t_n}$ (see Line 15 and 7) as the sole summary of historical visual information, preventing linear visual KV cache growth and ensuring constant visual context complexity relative to sequence length.

Finally, to achieve overall constant complexity for arbitrarily long videos, the FlowNar-C variant additionally employs a last-$k$ strategy for the narration context $\mathcal{C}^{\text{nar}}$, retaining only the KV cache entries for the $k$ most recent narrations, conceptually mirroring the sliding window attention mechanism (Beltagy et al., 2020; Jiang et al., 2023).

## 3.5. Training

We train our model on long videos containing multiple consecutive segments (*e.g.*, $\mathcal{S}_{n-2}, \mathcal{S}_{n-1}, \mathcal{S}_n$). For each segment $\mathcal{S}_n$, its sequence of frame embeddings $\{\mathbf{X}_t\}_{t \in \mathcal{S}_n}$ is prepended with the memory tokens $\mathbf{M}_{t_{n-1}}$ computed at the end of the segment $\mathcal{S}_{n-1}$. When generating a narration for segment $\mathcal{S}_n$, we employ a specific attention mask (Fig. 4) to ensure that the model relies only on the current frame em-

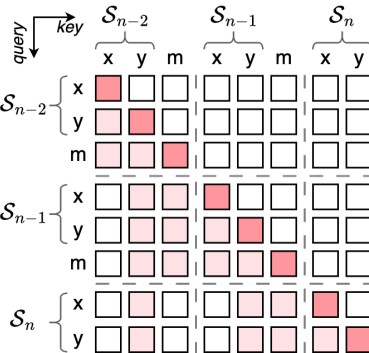

*Figure 4.* Training attention mask. Beyond standard causal masking, we explicitly block attention (white cells) to raw frames (x) and memories (m) from distant past segments. This enforces reliance on the most recent memory tokens and current frames for generating narrations (y), ensuring the model learns to utilize the compressed visual history as required during streaming.

beddings $\{\mathbf{X}_t\}_{t \in \mathcal{S}_n}$, the prepended memory $\mathbf{M}_{t_{n-1}}$ (as the summary of prior visual history), and previously generated narrations. In addition to the standard causal masking (upper triangle), this mask further restricts direct attention to raw frame tokens from preceding segments and memory tokens from distant past segments (white cells in the lower-left). The multi-segment training presents a different sequence structure to the LLM compared to inference, where the visual KV pairs are removed post-segment (Sec. 3.4). To reconcile positional IDs across these phases, we maintain a position counter to mimic training-style positional IDs during inference. We provide more details in Appendix B.3. FLOWNAR is trained end-to-end by optimizing a standard cross-entropy loss for next-token prediction. This loss is applied jointly to predict the ground-truth narration tokens ($y_n$) and the conditional [SKIP] token (Eq. 1).

## 4. Experiments

### 4.1. Experimental settings

**Datasets and data preparation.** We evaluate our method on three large-scale egocentric video datasets known for containing long-form recordings: Ego4D (Grauman et al., 2022), EgoExo4D (Grauman et al., 2024), and EpicKitchens100 (Damen et al., 2022). For all datasets, we leverage the official timestamped narrations and convert them to natural sentences (Chen et al., 2024) using Llama-8B (Meta, 2024) and GPT-4o (OpenAI, 2024). For all experiments, we adhere to the official training and validation splits provided by each dataset. Detailed dataset statistics, including the number of training/validation videos, video lengths, etc, are reported in Appendix B.4.

**Evaluation protocols and metrics.** We evaluate FLOWNAR on three key aspects: computational/memory efficiency, temporal alignment of narrations, and the textual quality of generated narrations. Efficiency is measured by Multiply-Accumulate operations (*MACs*) and peak GPU Memory (*VRAM*) for cached states during inference. To assess generation behavior, we use two protocol settings.

**Teacher-forcing protocol.** Following prior work (Chen et al., 2024), models are evaluated using *teacher-forcing*, where the *ground-truth* narration history is available at inference: the model generates the current narration $y_n$ conditioned on $\{y_j^{\text{gt}}\}_{j=1}^{n-1}$. Under this protocol, we report Perplexity (*PPL*) and *LM-Correctness* to assess token-level language modeling at each timestamp, and Time Difference (*TimeDiff*) and *Fluency* to evaluate both the language modeling and temporal effectiveness. These metrics are valid because model inputs and ground-truth labels are temporally aligned by construction.

**Self-conditioned protocol.** To simulate deployment, we evaluate models in a *self-conditioned* manner: each narration $y_n$ is conditioned only on the model's own previously generated narrations $\{y_j^{\text{pred}}\}_{j=1}^{n-1}$. Because predicted and ground-truth narration counts and boundaries generally differ, standard LM metrics under direct token-to-token alignment are not appropriate. We therefore use a *first-align-then-evaluate* pipeline that aligns predicted narrations to ground-truth segments before scoring. (1) For temporal alignment, we use segment-retrieval Precision, Recall, and F1, calculated by matching predicted and ground-truth segments via Intersection-over-Union (IoU, $\tau = 0.5$). (2) For narration quality, we retrieve for each ground-truth narration the best-matching predicted narration segment based on generalized IoU (GIoU) (Rezatofighi et al., 2019). We then compute standard captioning metrics CIDEr (C), METEOR (M), and ROUGE-L (R) on these matched pairs. Detailed metric definitions are provided in Appendix B.5.

**Baselines.** We compare against recent streaming narration models, namely Videollm-online (Chen et al., 2024), Videollm-mod (Wu et al., 2024), and LION-FS (Li et al., 2025b). Videollm-online and Videollm-mod are open-sourced. We retrained both using their official implementations to evaluate them under our self-conditioned protocol and ensure a fair comparison. All models use the same visual encoder (SigLIP-L/16 (Zhai et al., 2023)) and language model (Llama-3 (Meta, 2024)) in our experiments. For the teacher-forcing protocol, we report the performance numbers from the original papers. We provide more implementation details in Appendix B.6.

**Training cost.** Due to limited computing resources, we use Llama-3-1B as the language model by default, if not otherwise mentioned. Training FLOWNAR-1B on Ego4D requires 67 GPU-hours on 4×H100, compared to 36 GPU-hours for Videollm-online (~1.9×). This one-time overhead stems from the additional memory tokens and unoptimized

*Table 1.* Self-conditioned streaming narration benchmarks. With our dynamic context management (DCM), FLOWNAR variants consistently demonstrate significant improvements over the baselines.

| Dataset | Method | DCM | Efficiency | | Temporal Alignment | | | Narration Quality | | |
|---|---|---|---|---|---|---|---|---|---|---|
| | | | MACs↓ | Cache↓ | Prec.↑ | Rec.↑ | F1↑ | C↑ | M↑ | R↑ |
| Ego4D | Videollm-online | ✗ | 18.06T | 737.6M | 47.59 | 11.23 | 16.29 | 28.04 | 11.33 | 29.86 |
| | Videollm-mod | ✗ | **9.64**T | 482.6M | 45.74 | 10.80 | 14.75 | 27.01 | 10.78 | 28.78 |
| | FLOWNAR-C | ✓ | 16.50T | **20.2**M | 51.03 | 12.17 | 17.90 | 34.48 | 11.95 | 31.44 |
| | FLOWNAR | ✓ | 16.70T | 59.2M | **53.55** | **18.21** | **24.85** | **35.64** | **12.14** | **31.64** |
| Ego Exo4D | Videollm-online | ✗ | 22.45T | 878.5M | 50.96 | 26.36 | 31.77 | 69.88 | 23.53 | 50.19 |
| | Videollm-mod | ✗ | **13.21**T | 567.0M | 50.83 | 24.66 | 28.05 | 61.90 | 22.67 | 48.28 |
| | FLOWNAR-C | ✓ | 20.39T | **21.2**M | **52.48** | 26.69 | 32.82 | 71.64 | 24.31 | 50.72 |
| | FLOWNAR | ✓ | 21.12T | 125.9M | 51.40 | **27.28** | **32.99** | **75.33** | **24.46** | **51.29** |
| EK100 | Videollm-online | ✗ | 29.82T | 1096.0M | 51.07 | 8.74 | 12.98 | 29.00 | 14.23 | 45.26 |
| | Videollm-mod | ✗ | **15.99**T | 707.3M | 48.36 | 8.87 | 13.06 | 33.93 | 13.83 | 46.23 |
| | FLOWNAR-C | ✓ | 24.09T | **22.7**M | **52.71** | 19.21 | 25.20 | 37.28 | 14.39 | 45.41 |
| | FLOWNAR | ✓ | 24.42T | 65.3M | 49.12 | **23.12** | **29.12** | **46.63** | **14.93** | **46.25** |

attention kernels under our segment-level masking (Fig. 4); we view this as an implementation bottleneck that can be addressed in future work via custom kernels or sparse-attention adaptations (Sec. 4.5). This cost is amortized by the substantially lower inference cost (Fig. 1).

## 4.2. State-of-the-art comparison

**Self-conditioned streaming narration.** In Table 1, we evaluate narration performance using the realistic self-conditioned protocol. Videollm-mod demonstrates the lowest MACs and lower cache usage compared to Videollm-online. However, it still consistently caches key-values, resulting in a growing visual cache. We also note that Videollm-mod's routing mechanism appears vulnerable in this setting, where narration depends on self-generated, error-prone history, resulting in overall lower narration metrics than Videollm-online. Both FLOWNAR and FLOWNAR-C demonstrate markedly improved performance over the Videollm-online and Videollm-mod baselines. FLOWNAR consistently achieves the best temporal alignment F1 scores across all datasets, driven by significant gains in Recall. FLOWNAR-C, while offering greater efficiency, shows slight decreases in narration quality compared to FLOWNAR, highlighting the effectiveness of more extensive narration history. Notably, FLOWNAR-C reduces cache usage by up to 48.3× compared to Videollm-online on EK100. We attribute the significant performance gain to our robust dynamic context management (DCM) strategy. Baseline approaches risk compounding errors by caching extensive, potentially noisy Key-Value pairs from all past visual features and self-generated narrations. In contrast, FLOWNAR mitigates this by explicitly pruning the visual KV cache after each narration and relying on compact memory tokens to summarize visual history. This prevents conditioning on po-

*Table 2.* Teacher-forced narration benchmark on Ego4D. Our models achieve competitive metrics to state-of-the-art models.

| Method | Frame Strategy | Narration Quality | | | |
|---|---|---|---|---|---|
| | | PPL↓ | TimeDiff↓ | Fluency↑ | LM-Corr.↑ |
| Videollm-online | 1 | 2.430 | 2.320 | 42.6% | – |
| Videollm-online | 1+3×3 | 2.400 | 2.050 | 45.3% | 49.0% |
| Videollm-mod | | 2.410 | **2.040** | 45.2% | 48.9% |
| LION-FS | | 2.090 | 2.150 | 46.1% | 52.4% |
| FLOWNAR-C | | 2.006 | 2.219 | 46.1% | 53.5% |
| FLOWNAR | | **1.995** | 2.195 | **46.4%** | **53.9%** |

tentially misaligned detailed visual features from incorrectly narrated past segments, thereby reducing error propagation and enhancing robustness in the self-conditioned setting, which explains the superior performance.

**Teacher-forced narration.** Table 2 presents results on the Ego4D benchmark under the teacher-forcing protocol. Benchmark numbers for comparison methods are taken from the original papers. For a fair comparison, we use Llama-3-8B as the language model. Under teacher-forcing, FLOWNAR attains competitive narration quality and shows particular strength on language modeling metrics (*e.g.*, PPL and LM-Correctness). Notably, the performance gap narrows compared to the self-conditioned protocol (Table 1). This is expected, as conditioning on ground-truth history effectively eliminates the error propagation that baseline models suffer from in realistic settings.

## 4.3. Ablations and analysis

**Ablation on visual history with self-conditioned protocol.** As shown in Table 3, dynamic context management (DCM) is crucial for the self-conditioned protocol. When DCM is not applied (all past frames retained without pruning,

*Table 3.* Ablation on visual history for Ego4D under the self-conditioned protocol. Our CLAM module combined with DCM effectively summarizes history, outperforming other strategies.

| Past Frames | DCM | Past Narrations | Narration Quality | | |
|---|---|---|---|---|---|
| | | | C↑ | M↑ | R↑ |
| ✗ | ✓ | all | 30.40 | 11.36 | 30.54 |
| recent | ✓ | all | 30.16 | 11.42 | 30.59 |
| all | ✗ | all | 28.04 | 11.33 | 29.86 |
| CLAM | ✓ | all | **35.64** | **12.14** | **31.64** |

*Table 4.* Ablation on narration condition on Ego4D.

| Trigger strategy | Method | Temporal Alignment | | | Narration Quality | | |
|---|---|---|---|---|---|---|---|
| | | Prec.↑ | Rec.↑ | F1↑ | C↑ | M↑ | R↑ |
| Static | Videollm-online | 37.11 | 10.46 | 12.68 | 24.26 | 11.34 | 28.90 |
| | FLOWNAR | 35.84 | 15.68 | 16.78 | 30.28 | 11.69 | 30.38 |
| Dyn. | Videollm-online | 47.59 | 11.23 | 16.29 | 28.04 | 11.33 | 29.86 |
| | FLOWNAR | **53.55** | **18.21** | **24.85** | **35.64** | **12.14** | **31.64** |

row 3), performance significantly drops. While retaining all past frames offers maximal information, it accumulates noise from less relevant segments and compounds error propagation from imperfect self-generated narrations, both of which distract the model. With DCM, our CLAM (last row) substantially outperforms using no past visual frames (row 1) or only recent past frames (row 2), achieving the best temporal-aligned narration quality. This highlights that CLAM, combined with DCM's pruning of detailed past frame cache, provides a robust and effective visual history.

**Narration trigger condition.** We compare our dynamic two-threshold trigger (Sec. 3.2) against the static baseline in Table 4. For the dynamic approach, we set $\theta_{low}$ to 0.5 and the refractory period to the averaged segment duration. While FLOWNAR already outperforms Videollm-online using the static strategy, switching to the dynamic trigger strategy further boosts performance significantly, validating the dynamic approach's effectiveness for controlling narration cadence and improving overall performance. Ablation on specific trigger threshold values is provided in Appendix C.1, Table 12.

**Ablation memory design.** We compare our CLAM against alternative visual memory strategies that fulfills the constant per-step computation and memory cost properties (Sec. 3.3) in Table 5. Baselines include simple recent frame retention, online K-Means (Zhou et al., 2024), MovieChat (Song et al., 2024) (which iteratively merges tokens based on similarity), TokenMLP (Ryoo et al., 2021; 2023) (using an MLP for memory updates), and a refactored version of RetNet (Sun et al., 2023). To ensure a fair comparison of the memory mechanisms themselves, the number of memory tokens is kept the same for all methods. As shown, CLAM outperforms all other methods across all metrics, highlighting the

*Table 5.* Ablation on memory design on Ego4D.

| Method | PPL↓ | TimeDiff↓ | Fluency↑ | LM-Corr.↑ |
|---|---|---|---|---|
| recent | 2.115 | 2.257 | 45.0% | 51.7% |
| K-Means | 2.114 | 2.248 | 45.0% | 51.7% |
| MovieChat | 2.105 | 2.265 | 45.1% | 51.9% |
| TokenMLP | 2.127 | 2.274 | 44.6% | 51.3% |
| Refac. RetNet | 2.092 | 2.239 | 45.3% | 52.1% |
| CLAM | **2.086** | **2.237** | **45.4**% | **52.2**% |

*Table 6.* Performance for various video durations on EK100 under the teacher-forcing protocol.

| Dur. [min] | PPL↓ | TimeDiff↓ | Fluency↑ | LM-Corr.↑ |
|---|---|---|---|---|
| ≤ 2 | 2.254 | **1.749** | 39.0% | 52.6% |
| 2 − 4 | 2.337 | 3.057 | **43.1**% | 50.1% |
| 4 − 8 | **2.188** | 3.181 | 42.0% | **52.7**% |
| ≥ 8 | 2.279 | 3.326 | 42.0% | 52.4% |

effectiveness of our memory design compared to prior approaches. A detailed analysis of the memory designs, as well as an ablation on the impact of varying the number of memory tokens, can be found in Appendix B.2 and C.1.

**Performance for various video durations.** To analyze the performance for different video lengths, we grouped the videos of EK100 into four groups depending on the video length. To decouple CLAM's representational capacity from error propagation, we evaluate our approach under both protocols. Under teacher-forcing (Table 6), perplexity and fluency remain stable even for videos longer than 8 minutes, indicating that CLAM's fixed-size state has sufficient capacity for long-range dependencies given accurate history. Under self-conditioning (Table 7), we observe a gradual degradation with length, which we attribute to compounding errors in the model's own predictions rather than to a capacity limitation of CLAM. Additional analyses are provided in Appendix C.1.

### 4.4. Qualitative results

Consistent with the quantitative results in Table 1, we observe that FLOWNAR effectively reduces hallucinations and performs more robustly than the Videollm-online baseline model trained with full visual context. For instance, in

*Table 7.* Performance for various video durations on EK100 under the self-conditioned protocol.

| Duration [min] | Temporal Alignment | | | Narration Quality | | |
|---|---|---|---|---|---|---|
| | Prec.↑ | Rec.↑ | F1↑ | C↑ | M↑ | R↑ |
| ≤ 2 | **57.05** | **32.58** | **40.21** | **77.47** | **17.07** | **49.21** |
| 2 − 4 | 48.51 | 23.14 | 29.01 | 54.99 | 15.98 | 46.89 |
| 4 − 8 | 44.07 | 17.80 | 23.15 | 59.60 | 15.96 | 47.05 |
| ≥ 8 | 41.41 | 12.79 | 16.93 | 37.94 | 14.13 | 45.46 |

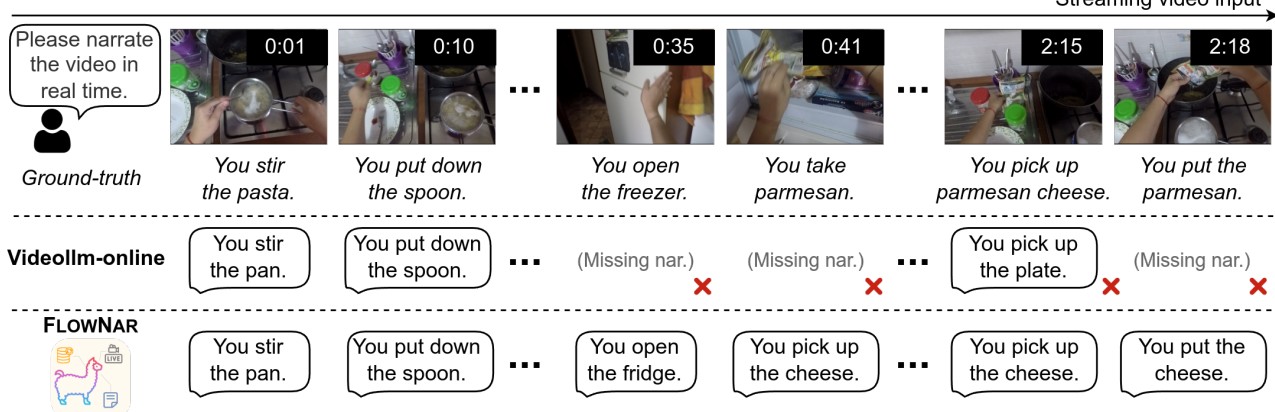

*Figure 5.* Examples of FLOWNAR on EpicKitchens100 (Damen et al., 2022). Text in red indicates incorrect narrations.

Fig. 5, our model correctly recognizes "open fridge" and "pick up the cheese" while the baseline fails to detect them or mistakenly identifies the cheese as plate. We provide several additional video demonstrations in Appendix A. We also evaluate zero-shot generalization on ActivityNet by applying our Ego4D-trained model directly to third-person videos without any fine-tuning. We find that despite the substantial domain shift from egocentric to exocentric views, FLOWNAR is able to identify complex actions. Corresponding qualitative results and failure cases are shown in Appendix C.5.

### 4.5. Limitations

While FLOWNAR achieves superior inference efficiency, the current training cost increases from 36 to 67 GPU hours compared to Videollm-online, due to additional memory tokens and unoptimized attention kernels under our segment-level attention masking (Fig. 4). We view this primarily as an implementation bottleneck. The training can be accelerated by developing custom kernels or adapt sparse attention mechanisms like NSA (Yuan et al., 2025) that are compatible with our masking constraints. While the three datasets (Ego4D, EgoExo4D, EK100) span diverse activity domains, video lengths, and vocabularies, our quantitative evaluation is confined to egocentric streaming narration. FLOWNAR works for third-person narration as well, but it needs to be trained on exocentric datasets to obtain better results as in the zero-shot setting (Appendix C.5). Extending FLOW-NAR to other streaming tasks such as streaming QA is an interesting direction for future work.

## 5. Conclusion

We introduced FLOWNAR, a novel framework for scalable streaming video narration. Extensive validation on benchmarks (Ego4D, EgoExo4D, EK100) demonstrates FLOW-NAR's effectiveness. It achieves comparable performance to state-of-the-art methods using oracle history and significantly outperforms them in realistic self-conditioned evaluations. This robust performance is enabled by our dynamic context management strategy, which combines visual cache pruning from completed segments with our Cross Linear Attentive Memory (CLAM) module for efficient, constant-cost visual history summarization. Consequently, FLOWNAR not only excels in narration quality but also achieves state-of-the-art streaming efficiency, supporting at least 10× longer videos at 3× higher FPS than prior online methods, demonstrating its suitability for long-form streaming narration.

## Acknowledgements

This work was supported by the JuBot project funded by the Carl-Zeiss-Foundation. The authors acknowledge support by the state of Baden-Württemberg through bwHPC. Experiments were performed on the HoreKa supercomputer funded by the Ministry of Science, Research and the Arts Baden-Württemberg and by the Federal Ministry of Education and Research. Juergen Gall has been supported by the ERC Consolidator Grant FORHUE (101044724).

## Impact Statement

Our efficient streaming framework contributes to developing low-resource assistive technologies, such as real-time narration for the visually impaired. While always-on video analysis carries potential privacy implications, our focus on efficient, bounded memory usage aims to make such systems more practical for beneficial edge-deployment applications.

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

## A. Online video demo

We include a diverse set of video examples that illustrate a range of behaviors produced by our model, including both effective and less effective outcomes. These examples are available online via a Google Drive link[1]. Audio is included using gTTS for demonstration purposes.

## B. Additional details

### B.1. Illustration of the self-conditioned generation process

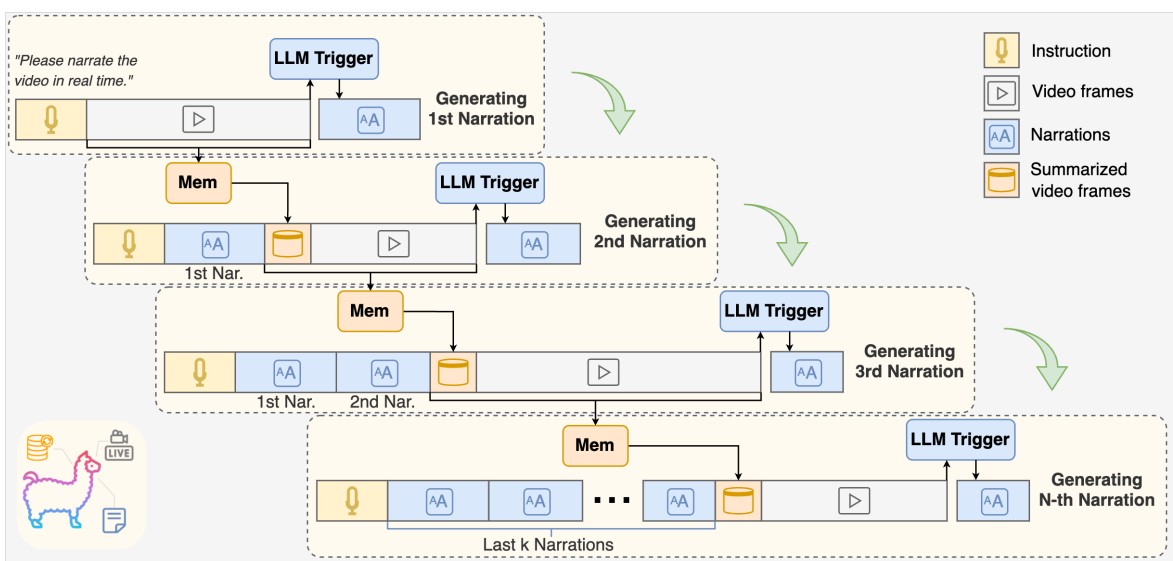

*Figure 6.* Illustration of the self-conditioned streaming narration process.

Figure 6 illustrates the self-conditioned streaming narration process (Sec. 3.4). Starting with an instruction, the LLM module evaluates incoming video frames alongside summarized visual history from the visual CLAM module (Mem) and its own previously generated narrations (blue) to decide when to narrate. Upon triggering, a new narration is generated and added to the history. This cycle repeats: the memory (Mem) continually updates its summary of processed visual frames, and the narration history accumulates. For extended sequences, narration history can be bounded (*e.g.*, using last-$k$ narrations), while the memory module consistently provides a compact summary of past visual information, ensuring bounded computation and memory cost for continuous generation.

### B.2. Memory design analysis

Two properties are essential for scalable streaming memory: (1) **bounded (effectively constant) memory complexity** so the visual state does not grow with time, and (2) **constant per-step computational complexity** so updates remain lightweight at each frame. Note that not all compact-token designs meet these requirements in practice. For example, a Q-Former (Li et al., 2023) can extract a fixed set of tokens at each time step, but applying cross-attention to summarize history at the next step typically requires storing past frame embeddings, causing the visual memory to grow and thus violating these properties. By contrast, linear-attention (Katharopoulos et al., 2020) formulations maintain a constant internal state that represents accumulated history, keeping memory bounded and enabling fast transitions between time steps, making them a natural candidate for refactoring into a streaming compression mechanism.

Baseline memory strategies in Table 5 differ in how they update a compact state. K-Means (Zhou et al., 2024) and MovieChat (Song et al., 2024) perform non-learned compression (clustering or token merging) and update memory using cosine similarity or Euclidean distance on raw frame embeddings. Since they lack learnable parameters and supervision, their updates cannot adapt to task signals. TokenMLP (Ryoo et al., 2023) introduces learnable parameters (MLPs) and benefits from end-to-end training, but it still effectively accumulates information from every incoming frame. When long,

[1] https://drive.google.com/drive/folders/18i6es_n1RwI4yHJ_6yxvt0MJuEdEa4DO

redundant segments dominate, the memory can become biased toward those segments and overlook rarer but important frames. Our CLAM design is intended to address these shortcomings. CLAM maintains a compact recurrent state and uses a per-token gating mechanism to adaptively control history retention, together with learnable readout queries to extract informative memory tokens. This combination enables CLAM to (a) bound visual memory, (b) keep per-step updates lightweight, and (c) reduce domination by redundant frames. We therefore attribute CLAM's superior empirical performance to these design choices.

CLAM belongs to the broader family of linear recurrent memories that maintain a fixed-size state $\mathbf{S}_t$ summarizing past inputs, alongside Mamba (Gu & Dao, 2023), GLA (Yang et al., 2024), and RetNet (Sun et al., 2023). Following the associative-memory analysis of Zhong et al. (2025), such a $D \times D$ state can store on the order of $O(D)$ independent key–value associations; we adopt this as a working bound rather than a formal guarantee on representable history. These designs differ along two axes that are relevant to streaming video: the *granularity of the gating mechanism* that controls how $\mathbf{S}_t$ is updated, and the *readout interface* used to expose the state to downstream consumers. On the gating axis, RetNet applies a fixed exponential decay, independent of input content; Mamba-2 uses a scalar-per-head input-dependent gate for hardware efficiency; CLAM, GLA, and Mamba-1 instead use a diagonal input-dependent gate, where each coordinate of $\mathbf{S}_t$ has its own input-dependent decay. The empirical advantage of CLAM over RetNet (Table 5) is consistent with this design choice. The second axis, the readout interface, is where CLAM is tailored to our setting. Rather than querying the state with the current token for next-token prediction as in language modeling, CLAM uses $M$ learnable query vectors as a cross-attention readout, producing a fixed-size set of memory tokens $\mathbf{M}_t = \mathbf{Q}\mathbf{S}_t$ that are prepended once per segment.

### B.3. Training-inference discrepancy regarding positional ids

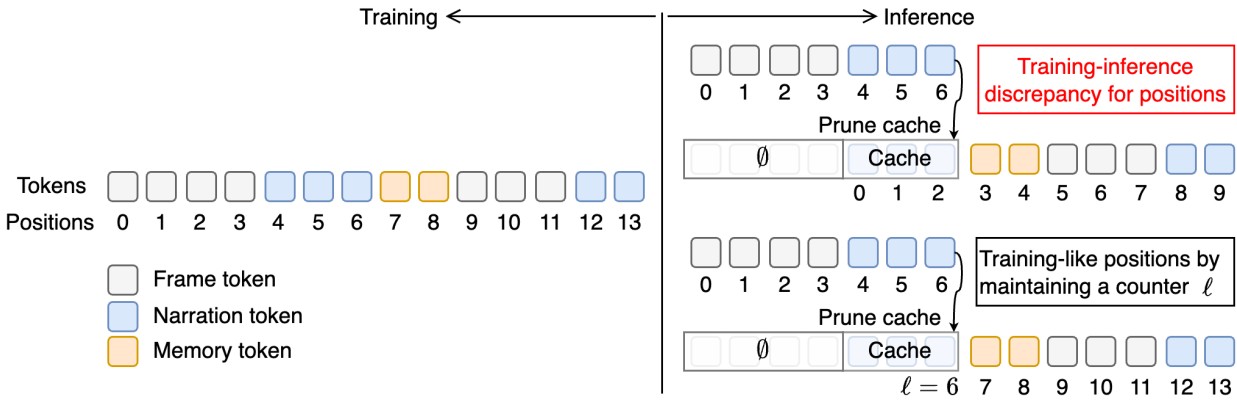

*Figure 7.* Discrepancy regarding positional ids between training and inference.

Figure 7 illustrates the training-inference discrepancy concerning positional ids in cached streaming models. During training (left), input tokens (frame, narration, memory) receive continuous positional ids (e.g., 0-13). However, naive inference (top-right) can erroneously reset these ids for new tokens after cache pruning, creating a mismatch with the training regime. To ensure consistency, our approach (bottom-right) employs a counter $\ell$ to assign continuous, training-like positional ids to incoming tokens (e.g., starting from $\ell = 6$ based on true sequence progression). This method preserves the validity of learned positional embeddings during streaming inference despite cache operations. A complete version of Alg. 1 with position counter $\ell$ can be found in Alg. 2.

### B.4. Dataset statistics

*Table 8.* Dataset statistics.

| Dataset | # Train | # Val | Video len. [s] | # Segments | Seg. dur. | # Nar. tokens |
|---|---|---|---|---|---|---|
| Ego4D | 102.986 | 17.059 | 237.7 ($\pm$92.0) | 53 | 4.5s | 12 |
| EgoExo4D | 3.219 | 826 | 151.4 ($\pm$232.7) | 58 | 2.6s | 17 |
| EpicKitchens100 | 495 | 138 | 493.9 ($\pm$621.3) | 112 | 4.4s | 10 |

**Algorithm 2** Self-conditioned streaming video narration generation with position counter

**Input:** Frame tokens $\{\mathbf{X}_t\}_{t=1}^T$, trigger threshold $\theta$
**Output:** Timestamped narrations $\Psi$

1: $(\Psi, \mathcal{C}_0^{\text{vid}}, \mathcal{C}_0^{\text{nar}}) \leftarrow (\emptyset, \emptyset, \emptyset)$          // Init output list & contexts
2: $n \leftarrow 1$; $b_{\text{new}} \leftarrow$ False          // Init segment index & flag
3: $\mathbf{S}_0 \leftarrow \mathbf{0}$; $\ell \leftarrow 0$          // Init memory state & position counter
4: **for** $t = 1$ **to** $T$ **do**
5:      $\mathbf{M}_t, \mathbf{S}_t \leftarrow \text{CLAM}(\mathbf{X}_t, \mathbf{S}_{t-1})$      // Update memory
6:      **if** $b_{\text{new}}$ **then**      // Prepend memory for new segment
7:          $\mathbf{X}_t \leftarrow [\mathbf{M}_{t_{n-1}} ; \mathbf{X}_t]$; $b_{\text{new}} \leftarrow$ False
8:      **end if**
9:      $\mathbf{E}_t \leftarrow \text{MLP}(\mathbf{X}_t)$      // Embedding alignment
10:     $\mathbf{pos}_t \leftarrow \ell + [1, \ldots, |\mathbf{E}_t|]$      // Training-style positions
11:     $\ell \leftarrow \ell + |\mathbf{E}_t|$      // Update position counter
12:     $p_{\texttt{[SKIP]}}, \mathcal{C}_t^{\text{vid}} \leftarrow \text{LLM}(\mathbf{E}_t, \mathbf{pos}_t, \mathcal{C}_{t-1}^{\text{vid}}, \mathcal{C}_{n-1}^{\text{nar}})$      // Process frame
13:     **if** $p_{\texttt{[SKIP]}} \leq \theta$ **then**      // Trigger condition met
14:        $t_n \leftarrow t$      // Update segment endpoint
15:        $y_n, \mathcal{C}_n^{\text{nar}}, \ell \leftarrow \text{LLM}(\mathcal{C}_{t_n}^{\text{vid}}, \mathcal{C}_{n-1}^{\text{nar}}, \ell)$      // Narration generation & position counter update
16:        $\Psi \leftarrow \Psi \cup \{(t, y_n)\}$      // Append result
17:        $\mathbf{M}_{t_n} \leftarrow \mathbf{M}_t$      // Update memory for new segment
18:        $\mathcal{C}_t^{\text{vid}} \leftarrow \emptyset$      // Discard visual context
19:        $n \leftarrow n + 1$; $b_{\text{new}} \leftarrow$ True
20:     **end if**
21: **end for**

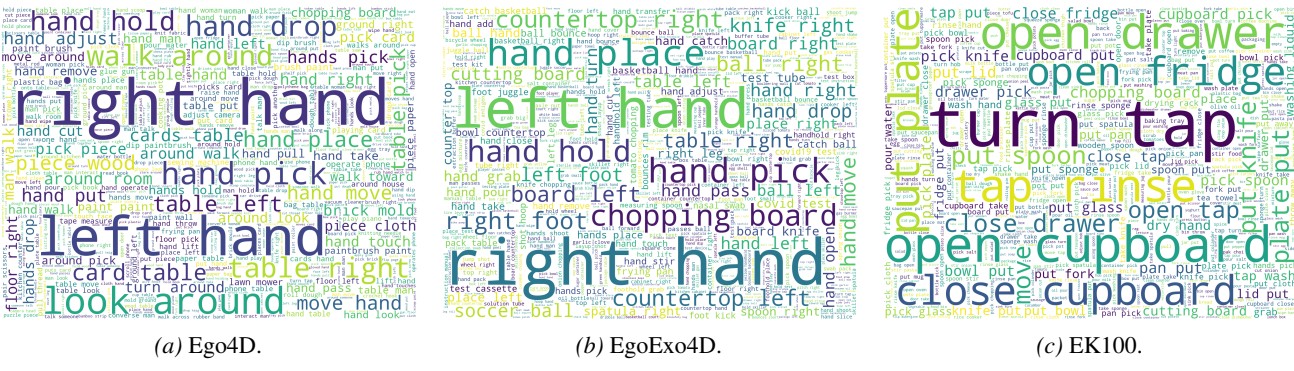

*(a) Ego4D.*          *(b) EgoExo4D.*          *(c) EK100.*

*Figure 8.* Wordcloud.

We conduct experiments on three challenging, long-form video datasets adapted for the streaming video narration task: Ego4D (Grauman et al., 2022), EgoExo4D (Grauman et al., 2024), and EpicKitchens100 (EK100) (Damen et al., 2022). Key statistics for these datasets, including the number of training/validation samples, average video length, number of segments per video, average segment duration, and average narration length in tokens (with Llama tokenizer), are summarized in Table 8. Figure 8 further illustrates the most frequent terms found in the narrations for each dataset.

### B.5. Protocols and metrics

Prior streaming narration evaluations (*e.g.*, Videollm-online, Videollm-mod, LION-FS) report quantitative results using ground-truth–conditioned interleaved token sequences constructed from labels (for example [vvnnvv...], where v = frame token and n = narration token). Under such a setup the model's input at each step is implicitly aligned with ground-truth narrations, so evaluation reduces to a next-token prediction task and language metrics can be computed by direct token-to-token matching. By contrast, in a realistic self-conditioned (*narration-when-triggered*) deployment the model conditions on its own previously generated narrations: predicted narrations may differ in number, timing, and boundaries from the ground

truth, making direct token-to-token alignment invalid. To enable fair, deployment-like assessment, we therefore adopt a *first-align-then-evaluate* strategy that aligns predictions to ground-truth segments post-hoc before applying metrics.

For an untrimmed video with ground truth timestamped narrations $\Psi_g = \{(t_n, y_n)\}_{n=1}^{N}$ and predicted timestamped narrations $\Psi_p = \{(t_{\hat{n}}, y_{\hat{n}})\}_{\hat{n}=1}^{\hat{N}}$, we evaluate three key aspects: temporal alignment, narration quality, and computational efficiency.

**Temporal alignment.** To assess the temporal alignment, we first convert timestamps into segments $S_g = \{s_n\}_{n=1}^{N}$ (ground truth) and $S_p = \{s_{\hat{n}}\}_{\hat{n}=1}^{\hat{N}}$ (predictions). We then compute pairwise temporal intersection-over-union (IoU) between all segments. Using an IoU threshold $\tau = 0.5$, segments are matched, and we calculate precision, recall, and F1 for the matched pairs:

$$\text{Precision}(\tau) = \frac{1}{\hat{N}} \sum_{\hat{n}=1}^{\hat{N}} \mathbb{I} \left[ \max_{1 \leq n \leq N} \text{IoU}(s_n, s_{\hat{n}}) \geq \tau \right] \tag{3}$$

$$\text{Recall}(\tau) = \frac{1}{N} \sum_{n=1}^{N} \mathbb{I} \left[ \max_{1 \leq \hat{n} \leq \hat{N}} \text{IoU}(s_n, s_{\hat{n}}) \geq \tau \right] \tag{4}$$

$$\text{F1}(\tau) = \frac{2 \cdot \text{Precision}(\tau) \cdot \text{Recall}(\tau)}{\text{Precision}(\tau) + \text{Recall}(\tau)}, \tag{5}$$

where $\mathbb{I}[\cdot]$ is the indicator function. Precision measures the fraction of predictions aligned with any ground truth, while recall measures the fraction of ground truths covered by predictions. Note that our formulation accommodates multiple annotators by allowing: (1) A prediction to match multiple ground truth segments (annotator diversity) (2) A ground truth segment to be covered by multiple predictions (redundant detection).

**Narration quality.** To assess the temporal-aligned narration quality, for each ground-truth segment, we find its best matching predicted segment using generalized intersection over union (GIoU) (Rezatofighi et al., 2019), and evaluate the predicted segment narration using standard captioning metrics, including n-gram-based metrics (e.g., CIDEr (Vedantam et al., 2015) and METEOR (Denkowski & Lavie, 2014)) and structural/semantic metrics such as ROUGE_L (Lin, 2004) (which measures longest common subsequences).

**Computational complexity.** Efficiency is critical for meeting the real-time demands of streaming video narration. We evaluate this using two metrics: video processing cost (measured in total multiply-accumulate operations, MACs) for generating narrations, and cache memory usage for storing intermediate results required in subsequent processing steps.

### B.6. Implementation details

All models were trained on $4\times$ NVIDIA H100 GPUs. The total training time of the 1B-parameter model on Ego4D is 67 GPU-hours, with a peak memory usage of 47 GB. We use a frozen SigLIP-L/16 (Zhai et al., 2023) as the visual encoder, processing frames at 2 FPS. Each frame is represented by $J$=10 tokens (1 CLS + 9 spatially averaged 3$\times$3 patch tokens). A 2-layer MLP projects these visual features from dimension $D = 1024$ to the LLM's hidden dimension $D_{\text{lm}} = 2048/4096$. For the language model, we employ Meta-Llama-3-1B/8B-Instruct (Meta, 2024), adapting all its linear layers with LoRA (Hu et al., 2022) (rank of 128, scaling factor of 256). Our CLAM module (Sec. 3.3, Fig. 3) uses $M = 20$ memory tokens and internally comprises one multi-head self-attention layer, our cross linear attention layer, and one feed-forward network. For training, we use the AdamW optimizer with a 2e-4 learning rate, an effective batch size of 64, and train for 2 epochs. A cosine learning rate scheduler with a 0.05 warm-up ratio and gradient clipping (max norm 1.0) are applied. For our dynamic two-threshold trigger (Sec. 3.2), $\theta_{\text{low}}$ is set to 0.5, with main $\theta$ set to 0.8, and the refractory period to 4 seconds, approximating the average segment duration across the datasets.

## C. Additional experiments

### C.1. Additional results

**Ablation on visual history with teacher-forcing protocol.** In Table 9, we compare different visual history strategies under the teacher-forced protocol, measuring both efficiency (MACs, Cache) and narration quality (PPL, TimeDiff). We observe that CLAM improves the narration quality for the teacher-forcing protocol (row 1 vs. row 4). Without CLAM, providing

*Table 9.* Ablation study on visual history with teacher-forcing protocol on Ego4D.

| Past Frames | Past Narr. | Efficiency | | Narration Quality | |
|---|---|---|---|---|---|
| | | MACs↓ | Cache↓ | PPL↓ | TimeDiff↓ |
| ✗ | all | **14.02**T | **57.5**M | 2.122 | 2.248 |
| recent | all | 15.68T | 58.8M | 2.115 | 2.257 |
| all | all | 18.06T | 737.6M | 2.118 | 2.245 |
| CLAM | all | 16.70T | 59.2M | **2.086** | **2.237** |

past visual context (rows 2 and 3) improves PPL only slightly.

*Table 10.* Ablation on training strategy for EgoExo4D and EK100. Pretraining on Ego4D is beneficial.

| Dataset | Finetune | PPL↓ | TimeDiff↓ | Fluency↑ | LM-Corr.↑ |
|---|---|---|---|---|---|
| Ego Exo4D | ✗ | 2.005 | 1.095 | 33.0% | 39.1% |
| | ✓ | **1.778** | **1.022** | **38.7%** | **46.4%** |
| EK100 | ✗ | 2.935 | 2.716 | 40.1% | 43.2% |
| | ✓ | **2.266** | **2.679** | **41.2%** | **51.9%** |

**Impact of training strategy for EgoExo4D and EK100.** As EgoExo4D and EK100 contain significantly fewer training videos compared to Ego4D, in Table 10, we investigate the impact of training strategy for these datasets. Specifically, we compare the performance of FLOWNAR trained from scratch vs. finetuned from an on Ego4D-pretrained checkpoint. Finetuning on the respective dataset leads to substantial improvements across all narration quality metrics. On EgoExo4D, finetuning improves LM-PPL from 2.005 to 1.778. Similarly, on EK100, finetuning results in a PPL improvement from 2.935 to 2.266. These results confirm the benefit of pretraining on Ego4D for EgoExo4D and EK100. We thus use this finetuning strategy for all models, including the baseline Videollm-online.

*Table 11.* Ablation on narration condition on EK100.

| Trigger strategy | Method | Temporal Alignment | | | Narration Quality | | |
|---|---|---|---|---|---|---|---|
| | | Prec.↑ | Rec.↑ | F1↑ | C↑ | M↑ | R↑ |
| Static | Videollm-online | 48.79 | 7.94 | 10.52 | 28.54 | 14.21 | 45.77 |
| | FLOWNAR | 38.39 | 19.39 | 18.04 | 37.33 | 14.66 | 45.26 |
| Dyn. | Videollm-online | **51.07** | 8.74 | 12.98 | 29.00 | 14.23 | 45.26 |
| | FLOWNAR | 49.12 | **23.12** | **29.12** | **46.63** | **14.93** | **46.25** |

**Narration trigger condition on EK100.** We compare our dynamic two-threshold trigger against the static baseline on EK100 in Table 11. For the dynamic approach, we set $\theta_{low}$ to 0.5 and the refractory period to 4s. Consistent with the results on Ego4D (Table 4), the dynamic trigger strategy boosts performance significantly.

**Impact of narration trigger threshold.** We analyze the impact of the narration trigger threshold $\theta$ for both static and dynamic strategies on EK100 in Table 12 and Figure 9. For our dynamic strategy, $\theta_{low}$ is set to 0.5 and the refractory period to 4 seconds. Table 12 shows that for the static strategy, $\theta = 0.7$ yields the best F1 score (18.04), though narration quality metrics vary. Our dynamic strategy with $\theta = 0.8$ significantly outperforms all static settings, achieving a much higher F1 (29.12) and superior narration quality (*e.g.*, CIDEr 46.63 vs. 37.33 for static $\theta = 0.7$). Figure 9 illustrates the effect on narration frequency. The static trigger leads to a very high number of narrations (and thus very short segments) at high $\theta$ values, while our dynamic strategy maintains a more stable and moderate narration frequency and segment duration.

**Ablation on narration history.** As shown in Table 13, adding our CLAM to a minimal baseline (row 2 vs. 1) offers an initial performance gain for the teacher-forcing protocol. Subsequently incorporating narration history (rows 3-5) yields further quality boost. While longer narration history ($k$) improves quality at a slight cost increase, we select $k = 10$ (last row) for our FLOWNAR-C configuration as it provides the best balance and overall performance in this efficient setting.

*Table 12.* Ablation on trigger threshold on EK100.

| Strategy | $\theta$ | Temporal Alignment | | | Narration Quality | | |
|---|---|---|---|---|---|---|---|
| | | Prec.↑ | Rec.↑ | F1↑ | C↑ | M↑ | R↑ |
| Static | 0.8 | 5.24 | 20.16 | 7.38 | 33.00 | 14.59 | 44.50 |
| | 0.7 | 38.39 | 19.39 | 18.04 | 37.33 | 14.66 | 45.26 |
| | 0.6 | 49.34 | 8.00 | 11.38 | 34.50 | 14.15 | 45.84 |
| | 0.4 | 47.65 | 5.10 | 8.03 | 36.02 | 14.04 | 46.09 |
| Dyn. | 0.8 | **49.12** | **23.12** | **29.12** | **46.63** | **14.93** | **46.25** |
| | 0.7 | 50.42 | 8.30 | 12.32 | 37.58 | 14.19 | 45.91 |

*Figure 9.* Narration frequency.

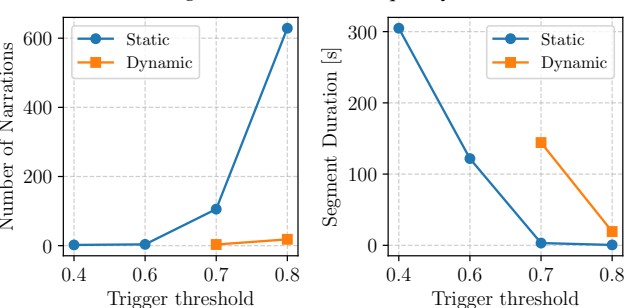

*Table 13.* Ablation on narration history on Ego4D.

| Past Frames | Past Narr. | Efficiency | | Narration Quality | |
|---|---|---|---|---|---|
| | | MACs↓ | Cache↓ | PPL↓ | TimeDiff↓ |
| ✗ | ✗ | **12.55T** | **11.4M** | 2.182 | 2.289 |
| CLAM | ✗ | 16.42T | 13.0M | 2.168 | 2.278 |
| CLAM | last-3 | 16.44T | 15.2M | 2.150 | 2.271 |
| CLAM | last-7 | 16.47T | 18.1M | 2.136 | 2.264 |
| CLAM | last-10 | 16.50T | 20.2M | **2.111** | **2.245** |

*Table 14.* Ablation on number of memory tokens on Ego4D.

| # Mem. | MACs↓ | Cache↓ | PPL↓ | Fluency↑ |
|---|---|---|---|---|
| 10 | **15.88T** | **58.5M** | 2.094 | 45.2% |
| 20 | 16.70T | 59.2M | **2.086** | **45.4%** |
| 30 | 17.52T | 59.9M | 2.099 | 45.2% |
| 50 | 19.15T | 61.3M | 2.098 | 45.3% |

**Ablation on number of memory tokens.** Table 14 shows an ablation on the number of memory tokens $M$ for CLAM on Ego4D. Increasing $M$ from 10 to 20 yields a small gain in narration quality (PPL 2.094 → 2.086; Fluency 45.2%→45.4%), but further increases to $M = 30$ or 50 produce no meaningful improvements while steadily raising compute (MACs), cache size, and training time due to longer sequences. Crucially, this behavior is expected: CLAM's storage capacity is governed by the recurrent state $\mathbf{S}_t \in \mathbb{R}^{D \times D}$, which accumulates and compresses past visual information, whereas the $M$ memory tokens are *readout* vectors computed from $\mathbf{S}_t$ and primarily control the expressivity of the readout step rather than the amount of stored history. Thus, increasing $M$ expands readout dimensionality but does not, by itself, increase the internal history capacity held in $\mathbf{S}_t$. For these reasons, and because $M = 20$ offers the best trade-off between modest quality gains and efficiency, we use $M = 20$ as our default.

*Table 15.* Effect of CLAM on narrations on Ego4D.

| Past Frames | Past Narr. | Narration Quality | | | |
|---|---|---|---|---|---|
| | | PPL↓ | TimeDiff↓ | Fluency↑ | LM-Corr.↑ |
| CLAM | CLAM | 2.174 | 2.275 | 44.0% | 50.2% |
| CLAM | last-10 | **2.111** | **2.245** | **45.0%** | **51.7%** |

**Effect of CLAM on narration history.** We investigate whether our CLAM module, designed for visual history compression, is also suitable for managing textual narration history (Table 15). When using CLAM to summarize past narrations instead of a standard last-$k$ strategy, we observe a notable degradation in narration quality across all metrics. This suggests CLAM is less effective for compressing textual sequences compared to its visual application. Two potential reasons include: (1) narration tokens are already highly condensed information, and further compression by CLAM might disrupt semantic integrity; and (2) the strict sequential order crucial for language may not be optimally preserved or leveraged by CLAM's current memory token representation. Thus, we retain a last-$k$ strategy for narration history in FLOWNAR-C.

**Impact of model size.** We compare FLOWNAR at two scales using the teacher-forcing protocol on Ego4D and EK100 in Table 16. The larger (8B) models yield modest but consistent gains, indicating that the 1B models perform well for many practical settings while the 8B models offer higher accuracy when compute resources are less constrained.

**Object hallucination frequency.** Our narration quality metrics (CIDEr, METEOR, ROUGE-L in Table 1) implicitly capture

*Table 16.* Effect of model scale on FLOWNAR.

| Dataset | Size | Narration Quality | | | |
| --- | --- | --- | --- | --- | --- |
| | | PPL↓ | TimeDiff↓ | Fluency↑ | LM-Corr.↑ |
| Ego4D | 1B | 2.086 | 2.237 | 45.4% | 52.2% |
| | 8B | **1.995** | **2.195** | **46.4%** | **53.9%** |
| EK100 | 1B | 2.266 | 2.679 | 41.2% | 51.9% |
| | 8B | **2.105** | **2.635** | **41.5%** | **53.8%** |

hallucination rates, as hallucinated content reduces overlap with ground truth. FLOWNAR consistently outperforms baselines across all three datasets. To provide more explicit evidence, we conducted an additional object-level accuracy analysis on EgoExo4D: we temporally align predicted and ground-truth segments (as in our self-conditioned protocol), extract active objects from both using Qwen3-8B, and compute semantic match rates. FLOWNAR achieves 36.6% object accuracy, compared to 33.8% for Videollm-online and 25.8% for Videollm-mod, representing an 8% relative improvement over the strongest baseline.

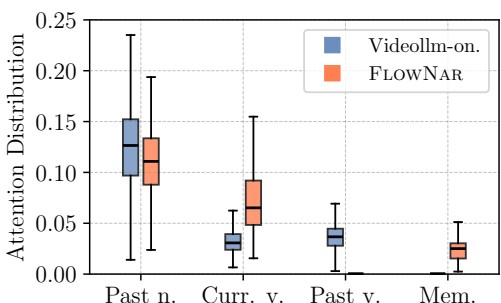

*Figure 10.* Attention value distribution.

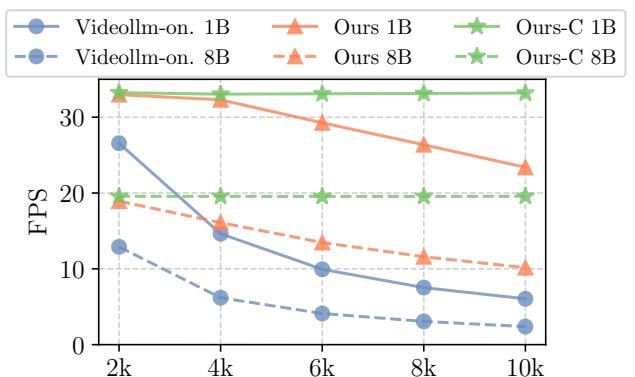

*Figure 11.* Speed comparison over sequence length. Our models maintain higher, more stable FPS.

## C.2. Attention analysis

Figure 10 illustrates the LLM's attention value distribution across different context components, comparing FLOWNAR with the Videollm-online baseline on the Ego4D validation set. The attention distributions are extracted from the last layer of respective models, averaged across all attention heads. These attentions are specifically extracted when generating a new narration. We show four categories of input tokens: past narrations (Past n.), current frame tokens (Curr. v.), past frame tokens (Past v.), and memory tokens (Mem.), while excluding other types of tokens such as instruction tokens for a clear analysis. For each category, we sum the attention values for all tokens. Both models significantly attend to past narration (Past n.), highlighting its importance for contextual understanding. A key difference emerges in how visual context is utilized: Videollm-online distributes some attention to raw past visual frames (Past v.). In contrast, FLOWNAR, due to its dynamic context management, shows no attention to raw past visual frames, instead utilizing its dedicated memory tokens (Mem.) to access historical visual information. With this memory mechanism in place, FLOWNAR also directs more attention to current visual frames (Curr. v.) compared to the baseline.

## C.3. Inference speed

Figure 11 illustrates the processing speed in Frames Per Second (FPS) against the number of processed frames, comparing FLOWNAR and FLOWNAR-C against the Videollm-online baseline (Chen et al., 2024) for both 1B and 8B language model sizes. The baseline shows a significant drop in FPS as more frames are processed, particularly for the 8B model, due to extensively accumulated context. By employing our dynamic context management strategy to remove historical visual context, FLOWNAR maintains a relatively stable FPS even after 10,000 frames ( 10 FPS for 8B, 23 FPS for 1B), greatly

exceeding the baseline. Furthermore, by strictly managing both visual and narration context for a constant memory footprint, FLOWNAR-C maintains a nearly constant processing speed irrespective of the number of frames evaluated, making it well-suited for extremely long-duration video processing.

## C.4. Video summary results

*Table 17.* Video summary results on Ego4D.

| Method | Finetuned | CIDEr ↑ | METEOR ↑ | ROUGE_L ↑ |
|---|---|---|---|---|
| LaVila (Zhao et al., 2023)+GPT2 | ✓ | 38.22 | 16.58 | 38.10 |
| LaVila (Zhao et al., 2023)+FLANT5 | ✓ | 39.13 | 16.88 | 38.77 |
| LaVila (Zhao et al., 2023) | ✓ | 24.63 | 15.30 | 33.31 |
| Video ReCap (Islam et al., 2024) | ✓ | **46.88** | **18.55** | **39.73** |
| BLIP2 (Li et al., 2023)+GPT3.5 | ✗ | 5.68 | 13.47 | 16.87 |
| LaVila (Zhao et al., 2023)+GPT3.5 | ✗ | 5.79 | 13.45 | 19.77 |
| FLOWNAR + Llama3 | ✗ | 11.20 | 17.94 | 32.63 |

To further assess the quality of the narrations generated by FLOWNAR in the self-conditioned setting, we used them as input to a Llama-3-8B model tasked with producing a concise video summary. The prompt provided to guide the Llama-3-8B model is detailed below.

```
You are an assistant trained to summarize timestamped human activity narrations.
The input is a list of tuples (timestep, narration) that may include repetitive
↪  or redundant entries due to being machine-generated.
Analyze the narrations to identify all distinct main activities (ignore
↪  minor/repetitive actions).
Condense the sequence by merging similar actions, eliminating redundancies, and
↪  preserving logical flow.
Format the summary as a sequence of actions, such as\n
[You were in a room, operated the laptop and tapped on the table.]\n
[You were in a workshop, picked a power woodcutter, then cut a wooden board.]\n
[You were in a kitchen, fried chapati in a frying pan and interacted with a
↪  woman.]\n\n
Now, please generate a concise summary for {content}.
```

Table 17 presents these video summary results. Notably, summaries generated from FLOWNAR's narrations achieve a METEOR score of 17.94 and a ROUGE_L score of 32.63. These scores are significantly higher than other non-finetuned methods (*e.g.*, LaVila+GPT3.5) and are competitive with, or even exceed, those from some finetuned baseline video summary methods (*e.g.*, LaVila for METEOR). This suggests that the autoregressively generated narrations from FLOWNAR are coherent and capture sufficient salient information from the long videos to enable effective downstream summary generation, even without task-specific finetuning for the narration model itself.

## C.5. More qualitative examples

**Additional qualitative results on ActivityNet.** We evaluate zero-shot generalization on ActivityNet by applying our Ego4D-trained model directly to third-person videos without any fine-tuning (Fig. 12). Despite the substantial domain shift from egocentric to exocentric views, FLOWNAR successfully identifies complex actions like painting a fence. We observe that because the model is trained on egocentric data (where the camera view dictates the actor), it grounds the narration to the subject currently dominating the visual focus. For instance, in Fig. 12b, the narration transitions from "You touch the watch" to "You play a game" as the camera cuts between different subjects. This confirms that the model's attention mechanism robustly tracks the salient action in the frame, correctly identifying fine-grained interactions even when applied zero-shot to third-person, multi-actor videos. We observe that the generated narrations retain the egocentric style, indicating that while visual semantics generalize robustly across viewpoints, the linguistic style sticks to the egocentric training data.

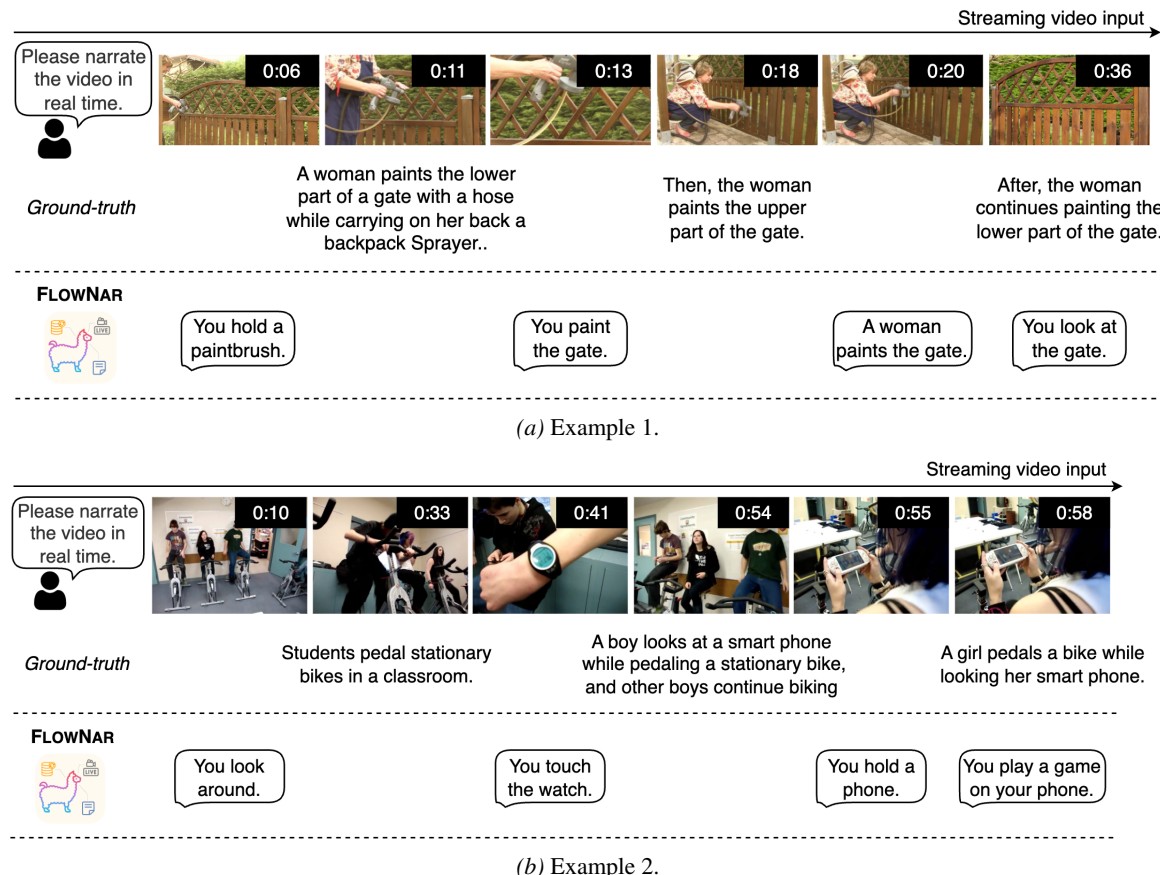

*(a) Example 1.*

*(b) Example 2.*

*Figure 12.* Zero-shot generalization to third-person video (ActivityNet). Despite the significant domain shift, the model correctly identifies key semantic activities (e.g., "paint the gate" and "play a game").

**Failure case analysis.** To understand the limitations of FLOWNAR, we analyze a representative failure case in Fig. 13. We observe that the model may hallucinate interactions with objects that are visible in the scene or semantically congruent with the environment (e.g., predicting "pick up the knife") even when they are not being actively manipulated. We attribute this to the aggressive spatial compression ($1$ CLS $+ 3 \times 3$ pooled tokens per frame) required to maintain real-time throughput. While efficient, this representation risks oversmoothing the fine-grained visual cues necessary to distinguish active hand-object interactions from background clutter. Furthermore, we observe that low-level visual degradations, such as the poor lighting at 0:34, can prevent the dynamic context management module from triggering a narration.

## D. More detailed related work

**Video captioning and streaming narration.** Research on generating textual descriptions from video began with video captioning, which focuses on producing a single sentence for short, trimmed clips (Venugopalan et al., 2015; Pan et al., 2017; Song et al., 2019; Wang et al., 2018a; Seo et al., 2022). To handle longer videos containing multiple events, the field progressed towards dense video captioning (Krishna et al., 2017; Zhou et al., 2018; Wang et al., 2018b; 2021a; Yang et al., 2023), which aims to localize temporal event segments and generate descriptions for each, and related tasks like video paragraph captioning (Lei et al., 2020) that generate more coherent multi-sentence descriptions. More recently, general-purpose vision-language models (VLMs) such as Qwen-VL (Bai et al., 2025) have demonstrated strong capabilities in understanding complex video content. However, these methods are primarily designed for offline processing, typically leveraging global context from complete video files. Concurrently, the AutoAD series (Han et al., 2023) has advanced the field of Movie Audio Description. However, these approaches typically operate in a clip-based manner and explicitly rely on subtitles or dialogue gaps to determine narration timing. Consequently, the majority of these methods operate *offline*, requiring access to the entire video, and often struggle to scale to long, unsegmented real-world recordings (Islam

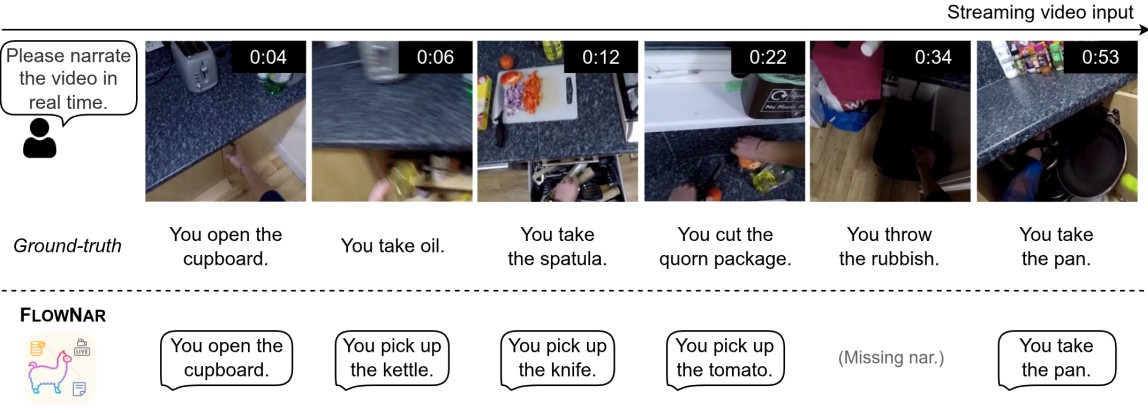

*Figure 13.* Qualitative failure analysis. We observe object hallucinations where the model predicts contextually plausible items rather than the active object, likely due to spatial downsampling ($3 \times 3$) oversmoothing fine details. Additionally, extreme lighting conditions (e.g., 0:34) can result in missed narrations.

et al., 2024). Addressing these limitations, the task of streaming video narration (Chen et al., 2024) was recently proposed, focusing on generating timely, timestamped descriptions continuously for incoming video segments in an online manner. We extend Chen et al. (2024) by supporting much longer videos and introducing a deployment-like self-conditioned evaluation and metrics.

**LMMs for online video understanding.** Large multimodal models (LMMs) (Alayrac et al., 2022; Li et al., 2023; Liu et al., 2023a; Zhu et al., 2024) have significantly advanced multimodal comprehension. Current LMMs address various video understanding benchmarks, including action recognition (Zhao et al., 2023; Qi et al., 2025), temporal action localization (Liu et al., 2024), and video dialogue/question answering (Li et al., 2025a; Song et al., 2024; Maaz et al., 2024; Zhang et al., 2023a; Lin et al., 2024). However, these models typically analyze entire videos *offline*, limiting their use in real-time applications like AR or autonomous driving. Consequently, benchmarks for online scenarios, such as action detection (De Geest et al., 2016; Wang et al., 2021b), localization (Kim et al., 2022), segmentation (Zhong et al., 2024), and anticipation (Furnari & Farinella, 2019; Zhong et al., 2023) using only past data, are increasingly critical. Videollm-online (Chen et al., 2024) represents the first LLM-based assistant for online video scenarios. Its efficient variant (Wu et al., 2024) reduces intermediate visual computation, supporting $1.7\times$ longer videos. LION-FS (Li et al., 2025b) introduces a fast/slow dual-pathway design for online video assistance. Despite these advances, scalability remains hindered as costs grow at least linearly with video length. In contrast, our method maintains relatively constant visual context complexity, enabling processing of significantly longer videos (empirically supporting at least $10\times$ greater lengths). ProVideLLM (Chatterjee et al., 2025) targets a different task: memory-efficient step recognition and forecasting over pre-segmented clips with a fixed observation window and a predefined action vocabulary, rather than autonomous open-vocabulary narration of an unsegmented stream. Dispider (Qian et al., 2025) and LiveCC (Chen et al., 2025) propose complementary approaches for real-time interaction and large-scale training, respectively. A separate line of streaming video LMMs targets *query-triggered* question answering rather than autonomous narration. ReKV (Di et al., 2025) stores all KV caches with RAM/disk offloading and retrieves relevant entries per incoming question; StreamMem (Yang et al., 2025) and StreamForest (Zeng et al., 2025) enforce bounded memory through attention-based KV pruning and event-level tree merging, respectively. Unlike these methods, which produce output only when an external query arrives, FLOWNAR must jointly decide *when* and *what* to narrate over a continuous stream without any prompting, requiring tight coupling between temporal localization and generation under bounded memory. We further note that streaming-style memory has also been explored in video generation (Qiu et al., 2025; An et al., 2025), where the goal is diffusion-based future-frame synthesis rather than describing observed frames.

**Dynamic context management in LLMs.** Efficiently managing the Key-Value (KV) cache is critical for autoregressive LLM inference over long sequences (Shi et al., 2024). Common strategies primarily address the textual KV cache. Sliding window attention (Beltagy et al., 2020; Jiang et al., 2023) offers simplicity by retaining only the KV pairs for a fixed window of recent tokens. More sophisticated cache eviction techniques aim to selectively discard less relevant KV pairs based on attention scores (Xiao et al., 2024; Han et al., 2024; Liu et al., 2023b; Zhang et al., 2023b; Adnan et al., 2024), or sparsification (Tang et al., 2025; Yao et al., 2024; Devoto et al., 2024), potentially preserving more crucial long-range information within a memory budget. Moving to the multi-modal regime, other works also address efficient visual history

management. For instance, MovieChat (Song et al., 2024) merges similar visual tokens, while Zhou et al. (2024) use online K-Means clustering for frame features. Different from these methods targeting textual caches or using alternative visual compression strategies, we introduce a neural memory mechanism specifically designed to efficiently manage and compress long-term visual context for streaming video analysis, and experimentally demonstrate its superiority.

