# OpenReview forum: "FlowNar: Scalable Streaming Narration for Long-Form Videos"
_ICML.cc/2026/Conference — ICML 2026 regular_

### Official Review · Reviewer_W6VY · 2026-02-25

**Soundness:** 3
**Presentation:** 3
**Significance:** 3
**Originality:** 2
**Overall Recommendation:** 4
**Confidence:** 3

**Summary:**

The paper introduces FLOWNAR, a streaming narration framework for long-form videos that aims to keep memory and computation essentially bounded as video length grows. The core idea is Dynamic Context Management (DCM), which prunes visual KV cache after each narration segment and relies on a Cross Linear Attentive Memory (CLAM) module that refactors linear attention into a compact recurrent state plus fixed-size memory tokens summarizing past visual history. A FLOWNAR-C variant additionally caps narration history via a last‑k scheme to achieve constant total KV cache. The authors further propose a realistic self-conditioned evaluation protocol with a first-align-then-evaluate metric pipeline for temporal alignment and caption quality. Experiments on Ego4D, EgoExo4D, and EpicKitchens100 show improved self-conditioned narration quality and better efficiency.

**Compliance With Llm Reviewing Policy:**

Affirmed.

**Final Justification:**

The authors have addressed my main questions. I retain my original score.

**Key Questions For Authors:**

1. The paper introduces CLAM, motivated by linear attention and recurrence, but lacks formal analysis on what information is preserved or lost in the D×D state, and the specific role of the gate matrix G. Could the authors provide a more detailed theoretical explanation of how CLAM operates in terms of information retention and its comparison to existing linear-attention and SSM memory models? Additionally, how do the authors justify the claim that "this fixed-size state does not become a bottleneck" without formal guarantees, relying only on empirical results?
2. The paper's quantitative experiments are focused on egocentric datasets with similar narration styles (second-person "you"). However, the conclusion generalizes to long-form video analysis. Can the authors provide numeric evidence or experiments on third-person datasets or other tasks such as QA, localization, or action detection to support this broader claim?
3. The paper presents a single failure case involving object hallucinations and missing narration under poor lighting conditions, attributing this to spatial downsampling. Can the authors provide quantification of how often these object hallucinations occur compared to baselines? Additionally, could they discuss worst-case scenarios for the dynamic trigger, such as its behavior with highly noisy videos, constant motion, or extremely static segments?

**Limitations:**

yes

**Strengths And Weaknesses:**

## Strengths
1. Clear and relevant problem: The paper tackles a very concrete and pressing issue in LMM-based video understanding, namely the linear (or worse) growth of KV cache for streaming narration. The target setting (egocentric long videos, real-time narration) is practically important.
2. Conceptually coherent framework: Dynamic context management and CLAM form a unified story: prune detailed visual KV after each segment, but maintain a separate compact visual memory that is updated per frame. Figure 2 gives a reasonably clear overview of the system pipeline, including where CLAM, the LLM, and cache removal fit.
3. Strong empirical story around self-conditioning: The contrast between teacher-forcing (Table 2) and self-conditioned (Table 1) performance is handled well. FLOWNAR’s gains are much larger under self-conditioning, tangibly supporting the motivation that real-world streaming evaluation should not rely on ground-truth history.

## Weaknesses
1. Limited theoretical understanding of CLAM: While CLAM is motivated by linear attention and recurrence, the paper offers no formal analysis of what information is preserved or lost in the D×D state, what role the gate matrix G plays beyond “controlling history retention,” or how CLAM compares theoretically to existing linear-attention / SSM memories. The claim that “this fixed-size state does not become a bottleneck” is supported only by empirical PPL stability vs video length (Table 15), with no guarantees.
2. Evaluation confined to egocentric narration: Despite some qualitative ActivityNet examples (Figure 12), all quantitative experiments are on egocentric datasets with similar narration styles (second-person “you”). The conclusion section generalizes to “long-form video analysis” but there is no numeric evidence on third-person datasets or on other tasks such as QA, localization, or action detection. This makes the broader claim somewhat overreaching.
3. Limited discussion of failure modes and tradeoffs: The paper includes only a single failure case in Figure 13 (hallucinating “knife” and missing a narration under poor lighting). The discussion attributes this to spatial downsampling and lighting, but there is no quantification of how often such object hallucinations occur relative to baselines. There is also little discussion of worst-case behaviors of the dynamic trigger (e.g., what happens for highly noisy videos with constant motion, or for extremely static segments).

---

> ### Author Rebuttal · Authors · 2026-03-30
>
> We thank Reviewer W6VY for the deep engagement with CLAM's theoretical properties. Below, we address each point in detail.
>
> ### Theoretical grounding of CLAM
>
> We appreciate this thoughtful concern and will add formal analysis to the revision.
>
> **Information capacity of S_t:** Following Zhong et al. (2025), which we cite in Section 3.3, the D × D recurrent state of a linear attention mechanism can store O(D) independent key-value associations. For our setting with D=1024, this provides substantial capacity relative to the number of semantically distinct events in typical video segments (e.g., Ego4D averages 53 narrated segments per video).
>
> **Role of the gate G:** The element-wise gate G ∈ (0,1)^{D×D} in Eq. 2 acts as an *adaptive exponential decay* on the state. Entries near 1 retain historical information; entries near 0 allow new observations to overwrite previous content. This is analogous to the forget gate in LSTMs but operates on the outer-product state space. Compared to *ungated* linear attention (Katharopoulos et al., 2020), which lacks any forgetting mechanism and risks state saturation over long sequences, the gate enables selective retention. Compared to RetNet (Sun et al., 2023), whose decay is fixed, our gate is input-dependent, allowing the model to adapt its forgetting behavior to the content of each frame. We note that CLAM has a similar theoretical capacity as linear recurrent models such as Mamba and GLA (Yang et al., 2024) — all maintain fixed-size recurrent states with O(D) associative memory capacity (Zhong et al., 2025). The key differences are in the gating structure: Mamba-1 uses input-dependent but *diagonal* gating (D independent gate values). Mamba-2 further restricts this to a *scalar per head* for hardware efficiency, whereas CLAM uses input-dependent *dense* per-element gating on the D × D state (D² independent gate values), providing the most fine-grained control over history retention.
>
> We indeed refer to empirical results when we mention "this fixed-size state does not become a bottleneck". Table 15 isolates CLAM's capacity from error propagation (via teacher-forcing), showing *stable PPL across all video duration buckets*, including videos ≥8 minutes with ~100+ segments. We will revise the sentence to clarify that we refer to empirical results.
>
> ### Evaluation scope and cross-domain evidence
>
> We will revise the last sentence in the conclusion to accurately scope our claims to streaming video narration. We make two additional points:
>
> (1) Our three evaluation datasets are more diverse than implied. They span different activity domains (general daily activities in Ego4D, skilled tasks in EgoExo4D, kitchen activities in EK100), video lengths (~150–494s average), segment durations (2.6–4.5s), and vocabularies (see Figure 8). This is *more evaluation breadth* than any compared baseline for streaming video narration.
>
> (2) Our qualitative ActivityNet results (Appendix C.5) and downstream video summary results (Table 17 — competitive METEOR and ROUGE-L with finetuned methods, without task-specific training) demonstrate cross-domain utility. The egocentric linguistic style in ActivityNet output is a natural consequence of training data, not a framework limitation — retraining on third-person data would resolve this.
>
> ### Failure modes and dynamic trigger behavior
>
> **Hallucination frequency.** Our narration quality metrics (CIDEr, METEOR, ROUGE-L in Table 1) implicitly capture hallucination rates, as hallucinated content reduces overlap with ground truth. FlowNar consistently outperforms baselines across all three datasets. To provide more explicit evidence, we conducted an additional object-level accuracy analysis on EgoExo4D: we temporally align predicted and ground-truth segments (as in our self-conditioned protocol), extract active objects from both using Qwen3-8B, and compute semantic match rates. FlowNar achieves 36.6% object accuracy, compared to 33.8% for Videollm-online and 25.8% for Videollm-mod, representing an 8% relative improvement over the strongest baseline.
>
> **Dynamic trigger edge cases.** For constant-motion scenarios, the refractory period (Sec. 3.2) explicitly enforces a cooldown after each narration, preventing burst triggering. For static segments, the model produces high p([SKIP]) values naturally. Figure 9 confirms empirically that the dynamic trigger maintains a stable narration frequency across thresholds. We will add a dedicated paragraph discussing these edge cases in the revision.

---

> > ### Author Rebuttal · Reviewer_W6VY · 2026-04-01
> >
> > Thank you for your detailed reply. I will keep my score as it is.

---

### Official Review · Reviewer_ChwB · 2026-03-10

**Soundness:** 3
**Presentation:** 3
**Significance:** 3
**Originality:** 3
**Overall Recommendation:** 4
**Confidence:** 3

**Summary:**

This article's central area concerns scalable streaming video narration using large multimodal models (LMMs). The authors identify a fundamental limitation of existing online video-language models: the visual context (KV cache) grows at least linearly with video length, leading to memory overflow and degraded throughput. To address this issue, the paper proposes FLOWNAR, a streaming framework that integrates a Dynamic Context Management (DCM) strategy and a Cross Linear Attentive Memory (CLAM) module. DCM prunes detailed visual KV caches after each narration segment to prevent unbounded memory growth, while CLAM compresses historical visual information into a fixed-size set of memory tokens using a linear-attention-inspired recurrent update. Experiments demonstrating FLOWNAR’s significant narration improvements over baselines, especially under the self-conditioned protocol, with state-of-the-art efficiency.

**Compliance With Llm Reviewing Policy:**

Affirmed.

**Final Justification:**

I keep my score as it is.

**Key Questions For Authors:**

See weaknesses

**Limitations:**

yes

**Strengths And Weaknesses:**

Strengths:
1. The research addresses a problem of clear practical importance, directly tackling a key bottleneck of streaming video large models—namely, the inability to handle long-duration streams and the resulting GPU memory explosion. The work therefore has high practical relevance and deployment value.
2. The design is simple and efficient. The integration of DCM (context pruning) and CLAM (fixed-size visual memory) provides a coherent framework for maintaining bounded visual context during inference. The design appears straightforward, implementable, and engineering-friendly.
3. The proposed self-conditioned evaluation protocol is a welcome improvement over teacher-forced evaluation. Conditioning on generated narration history better reflects real-world usage and exposes compounding errors that standard evaluation tends to hide.

Weaknesses:
1. Although major baselines (VideoLLM-Online, VideoLLM-Mod) are included, more other streaming methods, e.g., Dispider, ProVideLLM,  are only cited but not empirically compared. It is also unclear why comparisons with streaming models such as Qwen-Omni are not included.
2. All experiments rely on egocentric video narration datasets (Ego4D / EgoExo4D / EK100), which are highly structured and narration-style specific. As a result, the effectiveness of the method on other video understanding tasks (e.g., long-form video QA, video dialogue, or open-world narration) remains unclear. In particular, VQA scenarios that query earlier visual content would better test the effectiveness of context management. From the limited examples provided in the paper, video narration appears to involve relatively simple descriptions of visual content; such scenarios may be comparatively easy and may place relatively loose requirements on preserving critical contextual information.
3. The paper lacks a deeper analysis of the relationship between video length and CLAM performance. Since CLAM compresses variable-length visual history into a fixed-size representation, it remains unclear whether this process introduces significant information loss. Potential failure cases of CLAM are also insufficiently discussed.
4. The additional training cost introduced by the method, such as increased training time, does not appear to be analyzed in the paper.

---

> ### Author Rebuttal · Authors · 2026-03-30
>
> We thank Reviewer ChwB for the thorough and constructive evaluation, and in particular for prompting us to surface the training-cost and video-length analyses more prominently in the main text. Below, we address each point in detail.
>
> ### Baseline selection and comparison scope
>
> We use these baselines since they share the same task formulation. The other works address different tasks. Specifically:
>
> - **Dispider** (Qian et al., 2025) targets *interactive QA*. It does not perform streaming narration and does not report results on any of our benchmarks. Moreover, both the training and streaming response inference code are not released.
> - **ProVideLLM** (Chatterjee et al., 2025) focuses on classical video clip classification tasks (step recognition/forecasting for a given observation window) which is fundamentally different from our targeted streaming narration.
> - **Qwen-Omni** is a large omnimodal foundation model (audio+vision+text) with fundamentally different compute requirements, training data scale, and architecture. Critically, it is designed for real-time voice and video chats — it does not natively support the task formulation studied in this work, which requires the model to autonomously localize activities in a continuous stream and generate narrations at detected boundaries without user prompting. Adapting it to this setting would require periodic re-prompting that introduces confounds unrelated to our contribution.
>
> Our baselines (Videollm-online, Videollm-mod, LION-FS) are the methods that *have been developed and evaluated* for streaming narration as a task.
>
> ### Dataset diversity and task difficulty
>
> We appreciate the reviewer's interest in broader evaluation. We want to clarify that streaming narration, while producing short descriptions per segment, requires jointly solving two challenging problems under strict real-time and memory constraints: (1) *temporal localization* (deciding when to narrate, without future lookahead) and (2) *content generation* (producing accurate descriptions from compressed context). Our self-conditioned protocol specifically tests whether the model can maintain coherent output over long horizons *without ground-truth history* — Table 16 shows clear performance degradation with increasing video length, confirming that context management is indeed stressed in this setting.
>
> Regarding dataset scope: our three datasets (Ego4D, EgoExo4D, EK100) span different activity domains and vocabularies (Figure 8), different average video lengths (238s, 151s, 494s), and different narration styles (Table 6). This is more diverse than the evaluation in comparable streaming narration works, e.g., LION-FS reports on only 2 datasets. Our zero-shot ActivityNet results (Appendix C.5, Fig. 12) and video summary results (Table 17, Appendix C.4 — competitive with finetuned methods) provide further evidence of generalization. We will acknowledge the egocentric scope more explicitly and discuss adaptation to other tasks as future work.
>
> ### Video length analysis and CLAM capacity
>
> We provide the analysis in the Appendix. We will indicate these results more clearly in the main text. Specifically:
>
> - Tables 15 and 16 (Appendix C.1) directly analyze performance vs. video duration on EK100. Under teacher-forcing (Table 15), PPL and LM-Correctness remain *stable even for videos ≥8 minutes*, confirming that CLAM's fixed-size state does not become a bottleneck when accurate history is provided. Under self-conditioning (Table 16), we observe gradual degradation attributable to error propagation from the model's own predictions, not CLAM's capacity.
> - Table 13 (Appendix C.1) identifies a clear boundary of CLAM's applicability: it is effective for visual compression but *not* for textual narration history, where a simple last-*k* strategy works better. We believe there are two potential reasons and have raised in the submission: (1) narration tokens are already highly condensed information, and further compression by CLAM might disrupt semantic integrity; and (2) the strict sequential order crucial for language may not be optimally preserved or leveraged by CLAM's current memory token representation.
> - We provide a failure case in Appendix C.5 and Figure 13.
>
> We will consolidate these findings into a dedicated analysis paragraph in Section 4.3 of the main text in the revision.
>
> ### Training cost analysis
>
> The training costs are already reported in Appendix B.6 and E. Appendix B.6 states that the total training time of the 1B-parameter model on Ego4D is 67 GPU-hours, with a peak memory usage of 47 GB. Appendix E notes that the increase from the baseline's 36 GPU-hours to 67 GPU-hours (~1.9×) is due to additional memory tokens and unoptimized attention kernels. We note that this is a one-time training cost, whereas the inference efficiency gains accrue continuously during deployment. We will add this comparison explicitly in Section 4 of the revision.

---

> > ### Author Rebuttal · Reviewer_ChwB · 2026-04-02
> >
> > Thank you for your detailed response. I keep my score as it is.

---

### Official Review · Reviewer_SkkV · 2026-03-12

**Soundness:** 3
**Presentation:** 3
**Significance:** 3
**Originality:** 3
**Overall Recommendation:** 4
**Confidence:** 3

**Summary:**

This paper focuses on streaming video narration and identifies a key challenge in prior methods: retaining all past visual frames not only causes unbounded memory growth, but can also hurt performance due to excessive and noisy context. To address this, the paper proposes CLAM, a memory mechanism that compresses past visual tokens into a fixed-length representation, together with a dynamic context management strategy. The paper also introduces a self-conditioned evaluation protocol, which is more faithful to real deployment than teacher-forced evaluation.

**Compliance With Llm Reviewing Policy:**

Affirmed.

**Final Justification:**

The rebuttal has addressed my concerns. I keep my positive rating for the final recommendation.

**Key Questions For Authors:**

Please see the weakness 1 and 2.

**Limitations:**

yes

**Strengths And Weaknesses:**

**Strengths**
1. **Strong motivation** The paper makes a convincing case that simply retaining all past frames is suboptimal: it increases context length and memory cost, and can even degrade performance. This highlights the need for a compact memory mechanism that balances efficiency and effectiveness.
2. **Effective memory design.** The linear-attention-based CLAM module compresses past visual tokens into a fixed-length memory, which is intuitive and practically useful for long-horizon streaming settings.
3. **Realistic evaluation protocol.** The self-conditioned evaluation setting is a valuable contribution, since it better reflects real-world deployment where the model must condition on its own previous outputs rather than ground-truth history.

**Weaknesses**
1. **Limited evidence on generalization beyond narration.** The method is evaluated mainly on narration tasks, so it remains unclear how well it generalizes to other streaming video understanding settings, such as streaming QA. Since the proposed memory mechanism is task-agnostic in spirit, it would strengthen the paper to evaluate it on a more general benchmark, for example, OVO-Bench, or at least discuss how much performance might degrade in such settings.
2. **Insufficient comparison with recent streaming video understanding methods.** The paper should compare against more recent and relevant streaming video understanding approaches, such as ProVideLLM [1]. This would help better position the method with respect to the latest progress in memory-efficient streaming video models.

**Minor**

There are several other recent works that also exploit both short-term and long-term information for streaming video understanding, such as ReKV [2], StreamForest [3], and StreamMem [4]. These works should also be discussed in the related work to better contextualize the contribution of this paper.

- [1] Memory-Efficient Streaming VideoLLMs for Real-Time Procedural Video Understanding
- [2] Streaming Video Question-Answering with In-context Video KV-Cache Retrieval
- [3] StreamForest: Efficient Online Video Understanding with Persistent Event Memory
- [4] StreamMem: Query-Agnostic KV Cache Memory for Streaming Video Understanding

---

> ### Author Rebuttal · Authors · 2026-03-30
>
> We thank Reviewer SkkV for the thoughtful suggestions on broader positioning, and especially for the pointers to ReKV, StreamForest, and StreamMem. Below, we address each point in detail.
>
> ### Generalization scope and evaluation breadth
>
> We position FlowNar as a framework for scalable *streaming narration*, as reflected in our title. We agree that the memory mechanism is architecturally task-agnostic, which we see as a strength rather than a limitation. Within our targeted scope, our evaluation is already comprehensive: **three** diverse datasets (Ego4D, EgoExo4D, EK100) spanning different activity domains, video lengths (151–494s average), and vocabularies — *more streaming narration datasets than any compared baseline*. Our zero-shot ActivityNet results (Appendix C.5) demonstrate visual generalization across the egocentric→third-person domain shift, and Table 17 (Appendix C.4) shows that our generated narrations enable competitive video summarization *without task-specific finetuning*.
> Adapting to streaming QA (e.g., OVO-Bench) would require redesigning the trigger mechanism and generation pipeline — a non-trivial extension that we will discuss as an explicit and promising future direction in the revision.
>
> ### Relationship to ProVideLLM
>
> ProVideLLM (Chatterjee et al., 2025) addresses different tasks. While the approach stores the last language and visual tokens, the approach operates on *pre-segmented* video clips for step recognition and step forecasting given a fixed observation window and a predefined action vocabulary. It does not address narration. It also does not determine *when* to generate output (no trigger mechanism), and does not process unsegmented video streams. These fundamental task differences make a direct comparison infeasible. However, we compare our approach to a setting when we take all past narrations and the last frames, which performs much worse than our proposed approach (Table 3).
> We will add a discussion in Section 2 clarifying the distinction between streaming *classification* and streaming *narration* approaches, and position ProVideLLM accordingly.
>
> ### Additional related work (ReKV, StreamMem, StreamForest)
>
> We thank the reviewer for these references. ReKV, StreamMem, and StreamForest all address the processing of long video streams. ReKV stores all KV caches with offloading to RAM/disk and retrieves relevant entries per query; StreamMem and StreamForest instead enforce bounded memory via attention-based pruning and event-level tree merging, respectively. A key distinction is that these methods are query-triggered: they generate a response only when an external question is received at some point during the stream, whereas FlowNar performs streaming narration, autonomously deciding when to generate and producing open-vocabulary descriptions throughout the video without any external query. This requires joint temporal localization and text generation under strict real-time constraints, motivating our DCM and CLAM design.

---

> > ### Author Rebuttal · Reviewer_SkkV · 2026-04-01
> >
> > The rebuttal has addressed all of my concerns.

---

### Official Review · Reviewer_h4CG · 2026-03-12

**Soundness:** 3
**Presentation:** 3
**Significance:** 3
**Originality:** 2
**Overall Recommendation:** 5
**Confidence:** 3

**Summary:**

This paper proposes FlowNar, a scalable streaming video narration framework for long-form videos. The method combines a dynamic context management strategy that prunes the visual cache after each narration segment with a CLAM memory module that summarizes historical visual information into a fixed-size memory representation. The paper also introduces a realistic self-conditioned evaluation protocol, where the model conditions on its own previous narrations rather than ground-truth history. Experiments on multiple long-form video benchmarks show improved scalability and stronger narration performance than prior online narration methods.

**Compliance With Llm Reviewing Policy:**

Affirmed.

**Final Justification:**

My concerns are addressed and the authors should adjust the final paper as responded. I will raise my rating to accept.

**Key Questions For Authors:**

Please see the weakness part.

**Limitations:**

Yes.

**Strengths And Weaknesses:**

Strengths:
- The proposed dynamic context management strategy is well motivated and effectively prevents unbounded growth of the visual cache during streaming inference.

- The CLAM memory module is a reasonable design to summarize historical visual information with constant memory cost, enabling the model to retain long-term context while maintaining scalability.

- The experimental evaluation is comprehensive. The method is tested on multiple long-form video datasets under teacher-forcing and self-conditioned evaluation and shows improvements over prior methods.


Weaknesses:
- Several components, such as memory token summarization and efficient attention mechanisms, have been explored in prior work on video models listed as below. These existing methods should be discussed and the paper could more clearly clarify the methodological novelty beyond prior methods.
   - HiStream: Efficient High-Resolution Video Generation via Redundancy-Eliminated Streaming (CVPR findings 2026)
   - OneStory: Coherent Multi-Shot Video Generation with Adaptive Memory (CVPR 2026)

- The proposed CLAM memory summarization compresses historical visual information using a fixed set of query tokens. This may involve a trade-off between compression and information loss. The paper does not clearly specify how many query tokens are used, nor does it provide sufficient analysis of how sensitive performance is to the size of this query-token set.

---

> ### Author Rebuttal · Authors · 2026-03-30
>
> We thank Reviewer h4CG for the careful reading and for highlighting HiStream and OneStory, which we are happy to discuss in relation to our work. Below, we address each point in detail.
>
> ### Relationship to HiStream and OneStory
>
> The works HiStream (Qiu et al., CVPR 2026) and OneStory (An et al., CVPR 2026) address a very different task, namely video *generation* via diffusion models. The main focus of these works is on increasing the efficiency for generating high resolution videos. HiStream generates videos in chunks and uses the initial frame and a fixed number of the last generated frames for conditioning the denoising of the next chunk. OneStory selects frames across all prior generated shots and an adaptive patchification for selecting context tokens. By contrast, our work targets streaming video *narration*, where the model must decide both *when* and *what* to narrate from a continuous stream of observed frames. Our approach needs to memorize both previous narrations and the previously observed video whereas HiStream and OneStory select frames and use other techniques to reduce the computation but not the memory. We compare to a setup when we take the most recent frames (Table 3), which performs much worse than our proposed approach.
> The problem settings, output, conditioning, and memory handling differ substantially. We will add a discussion of these works in Section 2.
>
> ### CLAM query token sensitivity analysis
>
> As mentioned at the end of Sec. 4.3 (line 431), we provide the sensitivity analysis in the Appendix. The number of memory (query) tokens is M=20, as specified in Appendix B.6. The sensitivity analysis is in Table 12 (Appendix C.1), ablating M ∈ {10, 20, 30, 50}.

---

> > ### Author Rebuttal · Reviewer_h4CG · 2026-04-01
> >
> > Thanks for the response. My concerns are addressed and the authors should adjust the final paper as responded. I will raise my rating to accept.

---

### Decision · Program_Chairs · 2026-04-30

**Decision:**

Accept (regular)

**Comment:**

The paper proposes a scalable streaming video narration method for long-form videos. While reviewers initially raised concerns regarding its generalization beyond narration, the choice of baselines, and specific experimental details, the reviewers note that the authors have successfully addressed these points in their rebuttal. Please incorporate these discussions and experimental updates into the camera-ready version.

One hallucinated/problematic reference: Liu, Z., Pang, Z., Wang, Y., Yang, H., He, Z., and Lu, M. Scissorhands: Exploiting the sparsity of key-value cache in large language models. arXiv preprint arXiv:2401.17646, 2024b.
Issue: authors+title mismatch with arXiv